# PROVABLY EFFICIENT REINFORCEMENT LEARNING FOR ONLINE ADAPTIVE INFLUENCE MAXIMIZATION

## ABSTRACT

Online influence maximization aims to maximize the influence spread of a content in a social network with an unknown network model by selecting a few seed nodes. Recent studies followed a non-adaptive setting, where the seed nodes are selected before the start of the diffusion process and network parameters are updated when the diffusion stops. We consider an adaptive version of content-dependent online influence maximization problem where the seed nodes are sequentially activated based on real-time feedback. In this paper, we formulate the problem as an infinite-horizon discounted MDP under a linear diffusion process and present a model-based reinforcement learning solution. Our algorithm maintains a network model estimate and selects seed users adaptively, exploring the social network while improving the optimal policy optimistically. We establish $\widetilde{\mathcal{O}}(\sqrt{T})$ regret bound for our algorithm. Empirical evaluations on synthetic and real-world networks demonstrate the efficiency of our algorithm.

## 1 INTRODUCTION

Influence Maximization (IM) (Kempe et al., 2003; Kitsak et al., 2010; Centola & Macy, 2007), motivated by real-world social-network applications such as viral marketing, has been extensively studied in the past decades. In viral marketing, a marketer selects a set of users (seed nodes) with significant influence for content promotion. These selected users are expected to influence their social network neighbors, and such influence will be propagated across the network. With limited seed nodes, the goal of IM is to maximize the information spread over the network. A typical IM formulation models the social network as a directed graph and the associated edge weights are the propagation probabilities across users. Influence propagation is commonly modeled by a certain stochastic diffusion process, such as independent cascade (IC) model and linear threshold (LT) model (Kempe et al., 2003). A popular variant is topic-aware IM (Chen et al., 2015; 2016) where the activation probabilities are content-dependent and personalized, i.e., edge weights are different when propagating different contents.

Classical influence maximization solutions are studied in an offline setting, assuming activation probabilities are given (Kempe et al., 2003; Chen et al., 2009; 2010). However, this information may not be fully observable in many real-world applications. Online influence maximization (Chen et al., 2013; Wen et al., 2017; Vaswani et al., 2017) has recently attracted significant attention to tackle this problem, where an agent learns the activation probabilities by repeatedly interacting with the network. Most existing works on online influence maximization are formulated as a multi-armed bandits problem making a *non-adaptive* batch decision: at each round, the seed nodes are computed prior to the diffusion process by balancing exploring the unknown network and maximizing the influence spread; the agent observes either edge-level (Chen et al., 2013; Wen et al., 2017; Wu et al., 2019) or node-level (Vaswani et al., 2017; Li et al., 2020) activations when the diffusion finishes and updates its model. Combinatorial multi-armed bandits (Chen et al., 2013; Wang & Chen, 2017) and combinatorial linear bandits (Wen et al., 2017; Wu et al., 2019) algorithms have been proposed as solutions, where most works follow independent cascade model with edge-level feedback.

In contrast to the non-adaptive setting, adaptive influence maximization allows the agent to select seed nodes in a sequential manner after observing partial diffusion results (Golovin & Krause, 2011; Tong et al., 2016; Peng & Chen, 2019). The agent can achieve a higher influence spread since the decision adapts to the real-time feedback of diffusion. In viral marketing, the agent could observe

partial diffusion feedback from the customer and adjust the campaign for the rest of budgets based on current diffusion state. Unfortunately, online influence maximization in an adaptive setting is under-explored. Previous bandit-based solutions cannot be applied because the decisions of bandit algorithms are independent of the network state.

In this paper, we study the content-dependent *online adaptive influence maximization* problem: at each round, the agent selects a user-content pair to activate based on current network state, observes the immediate diffusion feedback, and updates its policy in real-time. The network's activation probabilities are content-dependent and are unknown to the agent. The agent's goal is to maximize the total influence spread. We formulate this problem as an infinite-horizon discounted Markov decision process (MDP), where the state is users' current activation status under different contents (user-content pairs), an action is to pick a user-content pair as the new seed, and the total reward is the discounted sum of active user counts. Specifically, we study the problem under the independent cascade model with *node-level* feedback. Similar to combinatorial linear bandits (Wen et al., 2017; Vaswani et al., 2017), we formulate a tensor network diffusion process where activation probabilities are assumed to be linear with respect to both user and content features. To tackle the problem of node-level feedback, we propose a Bernoulli independent cascade model, a linear approximation to the classic IC model which requires edge-level feedback to learn.

We propose a model-based reinforcement learning (RL) algorithm to learn the optimal adaptive policy. Our approach builds on prior work of bandit-based influence maximization algorithms (Chen et al., 2013; Wen et al., 2017; Wu et al., 2019) and has the following distinct features: (1) Our adaptive IM policy makes decisions and updates policy on the fly, without waiting till the end of diffusion process; (2) Our algorithm takes into consideration real-time feedback from the network, thus approaching a dynamic-optimal policy and outperforming bandit-based static-optimal solutions; (3) Our algorithm learns from node-level feedback, which greatly relaxes the common edge-level feedback assumption in previous works with IC model; (4) Our policy can handle content-dependent networks and select the best content for the right user for the campaign; (5) To improve computation efficiency, we adopt the slow switching strategy (Abbasi-Yadkori et al., 2011) that only update model parameter for $\mathcal{O}(d \log T)$ times, where $d$ is the feature space dimension. Our contributions are summarized as follows:

- We propose a linear tensor diffusion model for content propagation in social networks and formulate the problem as an infinite-horizon discounted MDP.
- We propose a tensor-regression-based RL influence maximization algorithm with optimistic planning that learns an adaptive policy from node-level feedback, which selects the content and next seed user based on current state of the network.
- We proved a $\widetilde{\mathcal{O}}(d\sqrt{T}/\Delta + \sqrt{dNKT})$[1] regret of our algorithm, where $T$ is the total rounds, $N$ is the number of users, $K$ is the number of contents, $\Delta$ is the coefficient for diffusion decay, $d$ is the dimension related to user and content feature. To our best knowledge, this is the first sublinear regret bound for online adaptive influence maximization.
- We empirically validated on synthetic and real-world social networks that our algorithm explores the unknown network more thoroughly than conventional bandit methods, achieving larger influence spread.

**Related Works.**  The classical works on (offline) influence maximization (Kempe et al., 2003; Chen et al., 2009; 2010) assume the network model, i.e., the activation probabilities, is known to the agent and the goal is to maximize the influence spread, i.e., total number of activated users. IM has been studied in a non-adaptive setting where the agent chooses the seed nodes before the diffusion starts (Kempe et al., 2003; Chen et al., 2009; 2010; Bourigault et al., 2016; Netrapalli & Sanghavi, 2012; Saito et al., 2008), or an adaptive setting where the agent sequentially selects the seed nodes adaptive to current diffusion results (Golovin & Krause, 2011; Tong et al., 2016; Han et al., 2018; Peng & Chen, 2019; Tong & Wang, 2020). Online influence maximization (Chen et al., 2013; 2015; Lei et al., 2015; Lugosi et al., 2019; Perrault et al., 2020; Zuo et al., 2022) is proposed to learn network model while selecting seed nodes in the non-adaptive setting. Existing works on online IM studies mostly follow IC model and edge-level feedback (Chen et al., 2013; Wang & Chen, 2017; Wen et al., 2017; Vaswani et al., 2017; Lugosi et al., 2019). Chen et al. (2013) and Wang & Chen (2017) formulated the online IM problem as combinatorial bandits problem and proposed combinatorial upper confidence

---

[1] $\widetilde{\mathcal{O}}(\cdot)$ ignores all logarithmic terms.

bound (CUCB) algorithm to estimate the activation probabilities of edges in a tabular manner. Wen et al. (2017) assumed a linear parameterization on each edge with known edge features and proposed a linear bandits-based solution. Our paper is the first to consider online influence maximization in the adaptive setting and formulate it as an RL problem. We also can handle the more challenging node-level feedback. Some recent works also explored settings beyond IC model and edge-level feedback. Li et al. (2020) studied online IM with linear threshold model, and proposed a linear bandits-based solution to model the linearity in LT model for node-level feedback. We also leveraged the linearity in diffusion model to handle node-level feedback similar to Li et al. (2020) but for IC model. Vaswani et al. (2017) considered diffusion model-independent setting using a heuristic objective function, but without theoretical guarantee of the heuristic. Olkhovskaya et al. (2018) studied UCB-based algorithm for node-level feedback, but their algorithm is designed only for certain random graph models such as stochastic block models and Chung–Lu models.

Our analysis is related to regret analysis of model-based reinforcement learning, which have been studied in various settings such as tabular MDP (Auer et al., 2008), linear/kernel MDP (Yang & Wang, 2020; Yang et al., 2020), factored MDP (Rosenberg & Mansour, 2021), general model class (Ayoub et al., 2020), etc. We provide a first problem-specific analysis for influence maximization. Our analysis differs from existing regret analysis in a couple of ways. First, although we focus on a linear model for network diffusion, the state-to-state transition of the IM is *highly nonlinear*, thus the value and Q functions for IM do not admit a linear model and invalidate linear/kernel MDP approaches. Second, due to the nature of network diffusion process, the state and its value can grow unboundedly for large networks, causing unbounded variance at the same time. Our analysis is specially tailored to such growth process over large networks and derive regret bound by focusing a high probability event where states stay bounded. To our best knowledge, this is a first IM-specific regret analysis for controlling unbounded growth process over large networks.

## 2 PROBLEM FORMULATION

We present a tensor network diffusion process to model user feature-dependent content feature-dependent network propagation. Our goal is to both select seed users and customize contents for influence maximization. Further, we formulate IM into an RL problem to enable much more delicate control of the network diffusion process based on real-time feedback.

### 2.1 TENSOR NETWORK DIFFUSION PROCESS

Consider a social network of $N$ users, where the network structure may be hidden. Let there be $K$ choices of contents. Let $s_{i,k} \in \{0, 1\}$ denote the status of an user-content pair, i.e., $s_{i,k} = 1$ if user $i$ is actively tweeting content $k$. The full state of the network is denoted by $s \in \{0, 1\}^{N \times K}$, a binary matrix. We focus on the **asymptotic regime of large networks**, i.e., $N$ can be arbitrarily large or even $N \to \infty$. We assume that each content can be propagated from one user to multiple users following an independent network diffusion process.

**Assumption 1** (Bernoulli Independent Cascade Model). *Let $s'$ be the next state. For each $k \in [K]$, we assume there is an underlying connectivity matrix $\mathbf{A}^k \in \mathbb{R}^{n \times n}$ such that*

$$\mathbb{P}(s'_{i,k} = 1|s) = \sum_{j \in [N]} \mathbf{A}^k_{i,j} s_{j,k}, \tag{1}$$

*And we assume $s'_{i,k}$'s are independent conditioned on $s$.*

Here $\mathbf{A}^k_{i,j}$ measures the level of influence user $j$ has over user $i$ for the $k$-th content. Therefore, the aggregate "influence" received by user $i$ is $\sum_j A^k_{i,j} s_{j,k}$. We model the status of user $i$ as a Bernoulli variable, which is parameterized by the aggregate "influence" received by user $i$.

Our model is closely related to the independent cascade model (Kempe et al., 2003). In IC model, the activation probability takes of the form $\mathbb{P}(s'_{i,k} = 1|s) = 1 - \prod_j(1 - \mathbf{A}^k_{i,j} s_{j,k})$. A limitation is that efficient estimation of IC model requires *edge-level* observations (Chen et al., 2013; Wang & Chen, 2017). Assumption 1 can be viewed as an linearized approximation to IC model, i.e., $1 - \prod_j(1 - \mathbf{A}^k_{i,j} s_{j,k}) \approx \sum_j \mathbf{A}^k_{i,j} s_{j,k}$ when all the $A$ values are tiny (see Assumption 3). In

Appendix F, we extend the assumption to the generalized linear setting and establish the regret bound for our algorithm.

Consider a parameterized network diffusion model based on user features and content features. Let the $i$-th user be associated with a user feature vector $x_i \in \mathbb{R}^{d_1}$, for all $i \in [N]$. Let the $k$-th content be associated with a content feature $\theta_k \in \mathbb{R}^{d_2}$, for all $k \in [K]$. We assume that the influence is linear with respect to both user and content feature.

**Assumption 2** (Content-Dependent Linear Tensor Model). *There exists a $d_1 \times d_1 \times d_2$ tensor $\mathcal{T}^* \in \mathbb{R}^{d_1 d_1 d_2}$ such that*

$$\mathbf{A}_{i,j}^k = \langle \mathcal{T}^*, x_i \otimes x_j \otimes \theta_k \rangle,$$

*where $\otimes$ denotes outer product and $\langle, \rangle$ denotes inner product.*

Note that this is different from the linear MDP model commonly studied in the theoretical RL literature (Jin et al., 2020). We focus on large networks where $N$ can be arbitrarily large. We also assume each individual user has bounded influence over its neighbors and the diffusion process has a natural decay property.

**Assumption 3** (Uniform transition probability upper bound). *There exists a constant $C > 0$ such that $\|\mathbf{A}\|_\infty \leq \frac{C}{NK}$.*

**Assumption 4** (Diffusion decay). *There exists $\Delta > 0$ such that $\sum_{i \in [N]} \mathbf{A}_{i,j}^k \leq 1 - \Delta$ for all $k, j$.*

Assumption 4 says that influence from any seed user has a discounting nature; without this assumption, some seed user may have infinite-long influence and make the diffusion process unbounded. This assumption also implies that, the "influence" of any seed user-content pair would last $\mathcal{O}(1/\Delta)$ time steps on average.

## 2.2 REINFORCEMENT LEARNING MODEL

We formulate the influence maximization problem as an infinite-horizon discounted MDP. Define the state space as $\mathcal{S} = \{0, 1\}^{N \times K}$ where 1 refers to activated user-content pair. At each timestep, the agent observes the current network state $s$ and picks an action $a \in \mathcal{A} := [N] \times [K]$ to activate one user-content pair. Let $s_a$ be the post-action state, i.e., $s_a = s + \mathbf{1}_a$. Then the state of network transitions following the network diffusion process, i.e., Assumptions 1,2. Since users are activated independently from one another, the state-transition law of the MDP admits a product structure:

$$\mathbb{P}(s'|s,a) = \prod_{i \in [N], k \in [K]} \mathbb{P}(s'_{i,k}|s,a).$$

At each state-action pair, the agent receives a reward $r(s,a) = \sum_{i,k} v_{i,k} s_{i,k}$ measuring the amount of influence over the network. For examples, if we let $v_{i,k} \equiv 1$, then we have $r(s,a) = \|s\|_1$, which counts the number of active users. Without lost of generality, we assume $v_{i,k} \leq 1$. Let $\pi : \mathcal{S} \mapsto \mathcal{A}$ be a decision policy. We measure the *value of policy $\pi$ at state $s$* as a cumulative sum of discounted rewards

$$V^\pi(s) = \mathbb{E}^\pi \left[ \sum_{t=1}^\infty \gamma^{t-1} r(s_t, a_t) \Big| s_1 = s \right].$$

Recall Assumption 4 that the influence of any action lasts $1/\Delta$ time steps. Thus, a natural choice of the discount factor to be $\gamma = 1 - o(\Delta)$. Finally, the policy optimization problem is to find $\pi^* = \arg\max_\pi V^\pi(s)$.

**Relation between Discounted MDP and Bandit IM model.** The discounted MDP formulation differs from the bandit IM optimization in two ways. (1) Our policy is dynamic and makes state-dependent decisions, while the bandit approach would make a batch of decisions only at the beginning of the diffusion process; (2) In both cases, the optimization objectives are sums of total influences from all seed users. The difference lies in how to measure the per-seed influence. In IM bandit, the per-seed influence is a cumulative sum calculated after the diffusion process is over. In our formulation, the per-seed influence is a cumulative $\gamma$-discounted sum of rewards from this seed's descendants. If we choose $\gamma = 1 - o(\Delta)$, these values differ by only $o(1)$ and we can make the difference arbitrarily small.

## 3 MODEL-BASED RL FOR INFLUENCE MAXIMIZATION (MORIMA)

To reduce the statistical complexity, we adopt a model-based RL approach for exploring the unknown network and learning the optimal policy. Our approach alternates between model estimation and policy update. Our algorithm calculates a bonus function based on the collected data and and add it to the reward, which dynamically trades-off between exploitation and exploration. We also adopt a slow switching technique to reduce computational burden.

**Tensor ridge regression for model estimate.** Under the linear tensor model (Assumption 2), we can use tensor ridge regression to perform model-based RL. This reduces the statistical complexity since the dimension of the unknown parameter is smaller. Furthermore, this approach only requires node-level feedback, While previous bandits approaches for IC model require edge-level feedback (Chen et al., 2013; Wen et al., 2017; Wu et al., 2019).

Specifically, let $s_a$ be the altered state after applying action $a$. Observe that, conditioned on $(s, a)$, the random variable $s'_{i,k}$ satisfies a linear relation:

$$\mathbb{E}[s'_{i,k}|s, a] = \sum_j \mathbf{A}^k_{i,j}(s_a)_{j,k} = \left\langle \mathcal{T}^*, x_i \otimes \left( \sum_j x_j \cdot (s_a)_{j,k} \right) \otimes \theta_k \right\rangle.$$

Denote for short $\phi_{i,k}(s, a) = x_i \otimes \left( \sum_j x_j \cdot (s_a)_{j,k} \right) \otimes \theta_k \in \mathbb{R}^{d_1 d_1 d_2}$, and $\phi^t_{i,k} = \phi_{i,k}(s_t, a_t)$. At time $t$, after observing the history $(s_1, a_1, \ldots, s_{t-1}, a_{t-1}, s_t)$, we estimate the tensor model by :

$$\widehat{\mathcal{T}}_t = \underset{\mathcal{T}}{\operatorname{argmin}} \sum_{\tau=1}^{t-1} \sum_{k=1}^{K} \sum_{i=1}^{N} (\langle \mathcal{T}, \phi^\tau_{i,k} \rangle - (s_{\tau+1})_{i,k})^2 + \lambda \|\mathcal{T}\|_2^2, \tag{2}$$

where $\|\mathcal{T}\|_2^2$ is calculated by vectorizing $\mathcal{T}$. This allows an analytical solution:

$$\widehat{\mathcal{T}}_t = \Sigma_{t-1}^{-1} B_{t-1}, \tag{3}$$

where

$$\Sigma_{t-1} = \lambda \mathbf{I} + \sum_{\tau=1}^{t-1} \sum_{k=1}^{K} \sum_{i=1}^{N} \phi^\tau_{i,k} \cdot (\phi^\tau_{i,k})^\top. \qquad B_{t-1} = \sum_{\tau=1}^{t-1} \sum_{k=1}^{K} \sum_{i=1}^{N} \phi^\tau_{i,k} \cdot (s_{\tau+1})_{i,k}. \tag{4}$$

Notice that the sizes of the covariance matrix $\Sigma_{t-1}$ and the right-hand-side term $B_{t-1}$ are $d \times d$ and $d$, respectively, where $d = d_1^2 d_2 \ll N$.

**Optimistic Planning with truncated-reward model.** To avoid the worst-case $O(NK)$ reward, we identify a high probability upper bound $\Lambda$ for the rewards and truncate the reward as $\widetilde{r}(s, a) = \min\{r(s, a), \Lambda\}$. Then based on the ridge regression estimation $\widehat{\mathcal{T}}_t$, we add a bonus term $b_t(s, a)$ to the truncated reward $\widetilde{r}$ and solve for an optimistic Q-function $Q^*_{\widehat{\mathcal{T}}_t, \widetilde{r}+b_t}(s, a)$ using the model estimate. Specifically, we can choose $\Lambda = \frac{6}{\Delta^2} \log(4NKT^3)$. For $\widehat{\mathcal{T}}_t$, we define the reward bonus as

$$b_t(s, a) = \frac{2\gamma\Lambda}{1 - \gamma} \sum_{i=1}^{N} \sum_{k=1}^{K} (1 \wedge \beta_t \cdot \|\phi_{i,k}(s, a)\|_{\Sigma_{t-1}^{-1}}), \tag{5}$$

where we use the notation $1 \wedge x = \min\{1, x\}$ and

$$\beta_t = \left( \frac{24}{\Delta} \sqrt{C_A/(NK) \cdot d \cdot \log(1 + NKL^2t/(d\lambda))} + 4 \right) \log(8N^2K^2t^2/\delta) + \sqrt{\lambda} \|\mathcal{T}^*\|_2 \tag{6}$$

with $L$ being an upper bound of $\|\phi^t_{i,k}\|_2$ and $d = d_1^2 d_2$.

This choice of $\beta_t$ ensures with high probability, $Q^*_{\widehat{\mathcal{T}}_t, \widetilde{r}+b_t}(s, a)$ is an upper bound of $\widetilde{Q}^*(s, a)$, which is the optimal Q-function for ground-truth transition with truncated-reward. We calculate the optimal truncated Q-function $Q^*_{\widehat{\mathcal{T}}_t, \widetilde{r}+b_t}(s, a)$ using value iteration with truncation (Algorithm 2).

**Slow switching.** To reduce computation overhead, we adopt a slow switching technique from bandit and RL literatures (Abbasi-Yadkori et al., 2011; Zhou et al., 2021b). The idea is that we only update model and policy when enough new data has been collected, via checking the covariance matrix. Specifically, say the most recent switching happens at time $t$, we choose to switch at time $t'$ only if

$$\det(\Sigma_{t'-1}) > 2\det(\Sigma_{t-1}).$$

After switching, we calculate the optimistic Q-function $Q_{t'} = Q^*_{\widehat{\mathcal{T}}_{t'}, \widetilde{r}+b_{t'}}(s, a)$. Then we pick actions greedily using $Q'_t$, i.e., $a = \operatorname{argmax}_a Q_{t'}(s, a)$, until the next switching.

---

**Algorithm 1** Model-based RL for Influence Maximization (MORIMA)

---

1: Initialize $\Sigma_1 = \lambda \boldsymbol{I}$, $B_1 = \boldsymbol{0}$. $Z = \det(\Sigma_1)$.
2: Calculate $\widehat{\mathcal{T}}_1$ and $b_1(s, a)$ and compute $Q_1 = Q^*_{\widehat{\mathcal{T}}_1, \widetilde{r}+b_1}(s, a)$.
3: Take the greedy action with respect to $Q_1$: $a_1 = \operatorname{argmax}_a Q_1(s_1, a)$.
4: **for** $t = 2, \cdots,$ **do**
5:     Calculate $\Sigma_{t-1}$ and $B_{t-1}$ according to Eqn. (4).
6:     **if** $\det(\Sigma_{t-1}) > 2Z$ **then**
7:         Calculate $\widehat{\mathcal{T}}_t$ and $b_t(s, a)$ according to Eqn. (3) and Eqn. (5).
8:         Compute the optimistic Q-function $Q_t = Q^*_{\widehat{\mathcal{T}}_t, \widetilde{r}+b_t}(s, a)$ (Algorithm 2).
9:         Set $Z = \det(\Sigma_{t-1})$.
10:     **else**
11:         Set $Q_t = Q_{t-1}$.
12:     **end if**
13:     Take the greedy action with respect to $Q_t$: $a_t = \operatorname{argmax}_a Q_t(s_t, a)$.
14: **end for**

---

**Algorithm 2** Truncated Value Iteration

---

1: **Input:** parameter $\mathcal{T}$, reward $\widetilde{r}(s, a)$, bonus term $b(s, a)$.
2: Initialize $Q(s, a) = \frac{\Lambda}{1-\gamma}$.
3: **while** Not Converged **do**
3:     $Q(s, a) \leftarrow \min\{\frac{\Lambda}{1-\gamma}, \widetilde{r}(s, a) + b(s, a) + \gamma \mathbb{E}_{s' \sim \mathbb{P}_{\mathcal{T}}(\cdot|s,a)} \max_{a'} Q(s', a')\}$
4: **end while**
5: **Return:** Optimistic Q function $Q(s, a)$.

---

**Full algorithm.** We put together the pieces and present the full Algorithm 1. The algorithm makes only $\mathcal{O}(d \log(T))$ model updates and policy updates until time $T$. Each model update can be done efficiently using least square regression. Policy updates require solving a new planning problem which can be combinatorially hard. In practice, one can solve the planning problem using approximate dynamic programming (Powell, 2007) or Monte-Carlo Tree Search (MCTS) methods (Browne et al., 2012), and we will use a two-step lookahead scheme in our experiments. For theoretical analysis, we assume access to a planning oracle that is able to find the optimal policy with respect to a known model $\widehat{\mathcal{T}}$. Relaxing such assumption to an approximated planning oracle can be also be done with minor algorithmic and analysis modifications.

## 4 REGRET ANALYSIS

In this section, we provide regret analysis for Algorithm 1. We define the regret for the infinite-horizon discounted MDP as in (Zhou et al., 2021b).

**Definition 1.** *For any possibly non-stationary policy $\pi$, the infinite-horizon discounted regret is defined as*

$$\text{Regret}(T) = \sum_{t=1}^{T} \Delta_t, \text{ where } \Delta_t = V^*(s_t) - V_t^\pi(s_t),$$

where $V^*$ is the optimal value function, and $V_t^\pi$ is defined as

$$V_t^\pi(s) = \mathbb{E}^\pi\Big[\sum_{i=0}^\infty \gamma^i r(s_{t+i}, a_{t+i})|s_1, \ldots, s_{t-1}, s_t = s\Big]$$

Now we present our main theorem.

**Theorem 2.** *Let Assumptions 1-4 hold. With probability at least $1 - \delta$, Algorithm 1 satisfies the following regret upper bound:*

$$\text{Regret}(T) \leq \widetilde{\mathcal{O}}\Big(\frac{1}{\Delta^2(1-\gamma)^2} \cdot \big(d\sqrt{C}/\Delta + \sqrt{dNK}\big) \cdot \sqrt{T}\Big) + \text{polylog}(T)\text{-terms},$$

*where $d = \dim(\mathcal{T}^*) = d_1^2 d_2$.*

We see that the dominant term of the regret is $\widetilde{\mathcal{O}}\Big(\frac{1}{\Delta^2(1-\gamma)^2} \cdot \big(d\sqrt{C}/\Delta + \sqrt{dNK}\big) \cdot \sqrt{T}\Big)$. Notice that the worst-case reward would scale with $NK$, while we managed to reduce the scaling of the regret to $1/\Delta^2$.

### 4.1 Proof sketch of Theorem 2

Next, we provide a proof sketch and defer the complete proof to Appendix C.

*Proof.* **High probability upper bounds for the size of active user-content pairs.** We utilize the diffusion decay assumption (Assumption 4) to provide a high probability upper bound on the number of active user-content pairs. We show that for any policy $\pi$, with probability at least $1 - p$, we have for all $t \geq 1$,

$$\|s_t\|_1 \leq \mathcal{O}(\frac{1}{\Delta^2} \log \frac{4t^2}{p}). \tag{7}$$

We see that although we have in total $NK$ user-content pairs, the number of active ones is constrained by a constant intrinsic to the network diffusion dynamics.

**Sharper bounds for the confidence region.** We derive a batched version of Bernstein-type self-normalized bounds from (Zhou et al., 2021b) and show that with high probability for all $t$, $\|\widehat{\mathcal{T}}_t - \mathcal{T}^*\|_{\Sigma_{t-1}} \leq \beta_t$, where $\beta_t$ can be chosen as $\widetilde{\mathcal{O}}(\sigma\sqrt{d} + 1)$ and $\sigma^2$ is the upper bound of $\text{var}[(s_{t+1})_{i,k}|s_t, a_t]$. Combing Eqn. (7) and Assumption 3, we have

$$\text{var}[(s_{t+1})_{i,k}|s_t, a_t] \leq \mathbb{E}[(s_{t+1})_{i,k}|s_t, a_t] = \sum_j \mathbf{A}_{i,j}^k(s_{ta_t})_{j,k} \leq \frac{C}{NK}(\|s_t\|_1 + 1) \leq \widetilde{\mathcal{O}}(\frac{C}{NK\Delta^2}).$$

Then $\beta_t = \widetilde{\mathcal{O}}(\sqrt{\frac{dC}{NK}}/\Delta + 1)$, which improves upon $\beta_t = \widetilde{\mathcal{O}}(\sqrt{d})$ given by the sub-Gaussian type self-normalized bounds.

**Surrogate regret of the truncated-reward model.** Since we essentially run our algorithm against the truncated-reward model, we define the surrogate regret as $\widetilde{\text{Regret}}(T) = \sum_{t=1}^T (\widetilde{V}^*(s_t) - \widetilde{V}_t^\pi(s_t))$, where $\widetilde{V}^*$ and $\widetilde{V}_t^\pi$ are computed using the truncated reward $\widetilde{r}(s, a) = \min\{r(s, a), \Lambda\}$. By Eqn. (7), with probability at least $1 - 1/(2NKT)$, under any policy, we have $r(s_t, a_t) \leq \|s_t\|_1 \leq \Lambda = \widetilde{\mathcal{O}}(1/\Delta^2)$ for all $t \leq T$. This means with high probability we have $r(s_t, a_t) = \widetilde{r}(s_t, a_t)$ and hence the true regret and the surrogate regret is similar. Specifically, we will show $\text{Regret}(T) \leq \widetilde{\text{Regret}}(T) + 1/(1 - \gamma)$.

**Bonus term.** Let $\mathbb{P}(s'|s, a)$ be the true transition probability and $\widehat{\mathbb{P}}(s'|s, a)$ be the empirical estimate. As a typical result in MDP theory, we require $b(s, a) \geq \gamma\|V\|_\infty \cdot \|\mathbb{P}(\cdot|s, a) - \widehat{\mathbb{P}}(\cdot|s, a)\|_1$ to ensure optimism. We exploit the fact that $\mathbb{P}(s'|s, a)$ and $\widehat{\mathbb{P}}(\cdot|s, a)$ is factorized, i.e., $\mathbb{P}(\cdot|s, a) = \otimes_{i=1}^N \otimes_{k=1}^K \mathbb{P}_{i,k}(\cdot|s, a)$ and $\widehat{\mathbb{P}}(\cdot|s, a) = \otimes_{i=1}^N \otimes_{k=1}^K \widehat{\mathbb{P}}_{i,k}(\cdot|s, a)$, which stem from the independence assumption (Assumption 1). This gives us $\|\mathbb{P}(\cdot|s, a) - \widehat{\mathbb{P}}(\cdot|s, a)\|_1 \leq \sum_{i=1}^N \sum_{k=1}^K \|\mathbb{P}_{i,k}(\cdot|s, a) - \widehat{\mathbb{P}}_{i,k}(\cdot|s, a)\|_1$. Notice that $\mathbb{P}_{i,k}(\cdot|s, a)$ is a Bernoulli distribution, then by Assumption 2 we have $\|\mathbb{P}_{i,k}(\cdot|s, a) - \widehat{\mathbb{P}}_{i,k}(\cdot|s, a)\|_1 \leq 2(1 \wedge |\langle \mathcal{T}^* - \widehat{\mathcal{T}}, \phi_{i,k}(s, a)\rangle|)$. Therefore, the bonus term can be chosen as Eqn. (5) and we ensure optimism at each time.

**Regret decomposition.** We have the following regret decomposition for the surrogate regret.

$$\widetilde{\text{Regret}}(T) \leq \mathcal{O}\Big\{ \frac{1}{1-\gamma}\Big[ \sum_{t=1}^{T} b_{t_s}(s_t, a_t) + \frac{2\gamma\Lambda}{1-\gamma}\sqrt{T\log\frac{1}{\delta}} + \Big(\frac{\Lambda}{1-\gamma}M\Big)\Big]\Big\},$$

where $M = M(T)$ is the total number of switches and we will show that $M = \widetilde{\mathcal{O}}(d)$. Then the dominant term of the regret is

$$\frac{1}{1-\gamma}\sum_{t=1}^{T} b_{t_s}(s_t, a_t) = \widetilde{\mathcal{O}}\Big(\frac{1}{(1-\gamma)^2\Delta^2}\Big) \cdot \beta_T \cdot \sum_{t=1}^{T}\sum_{i=1}^{N}\sum_{k=1}^{K}(1 \wedge \|\phi_{i,k}(s_t, a_t)\|_{\Sigma_{t_s-1}^{-1}})$$

$$\leq \widetilde{\mathcal{O}}\Big(\frac{1}{(1-\gamma)^2\Delta^2}\Big) \cdot \beta_T \cdot \widetilde{\mathcal{O}}(\sqrt{dNKT} + MNK).$$

where $t_s$ denotes the last switch up to time $t$, and the last inequality follows from a variant of Elliptical Potential Lemma. Plug in the choice of $\beta_T$ and we derive the result. □

### 4.2 EXTENSION TO GENERALIZED LINEAR INDEPENDENT CASCADE MODELS

In this subsection, we briefly show how to extend our algorithm and regret bound to generalized linear IC models. The details can be found in Appendix F.

**Assumption 5** (Generalized Linear Independent Cascade (GLIC) Model). *Let $s'$ be the next state. For each $k \in [K]$, we assume there is an underlying connectivity matrix $\mathbf{A}^k \in \mathbb{R}^{n \times n}$ such that*

$$\mathbb{P}(s'_{i,k} = 1|s) = \mu\Big( \sum_j \mathbf{A}^k_{i,j} s_{j,k} \Big), \tag{8}$$

*where $\mu : \mathbb{R} \to \mathbb{R}$ satisfies $\mu(0) = 0$ and $1/\kappa \leq \mu' \leq 1$ for some $\kappa \geq 1$. And we assume $s'_{i,k}$'s are independent conditioned on $s$.*

Given Assumption 5, there are two major differences extending MORIMA to GLIC models: (1) new objective of tensor model estimation under GLM, which does not have analytical solutions; and (2) new optimistic planning, which is analyzed following new tensor estimate. We have the following regret bound for MORIMA under GLIC.

**Theorem 3.** *Let Assumption 5, Assumptions 2-4 hold. With probability at least $1 - \delta$, Algorithm 1 satisfies the following regret upper bound:*

$$\text{Regret}(T) \leq \widetilde{\mathcal{O}}\Big( \frac{\kappa}{\Delta^2(1-\gamma)^2} \cdot \big(d\sqrt{C}/\Delta + \sqrt{dNK}\big) \cdot \sqrt{T} \Big) + \text{polylog}(T)\text{-terms},$$

*where $d = \dim(\mathcal{T}^*) = d_1^2 d_2$.*

## 5 EXPERIMENTS

We experiment with MORIMA (Algorithm 1) for influence maximization on both synthetic networks and a large-scale Twitter social network.

**Synthetic networks.** We run benchmark experiments on two synthetic networks. The first cascade network (see Appendix E Figure 2a), consisting of $N = 300$ users and $K = 4$ content types, is constructed to bear a hierarchical structure with users with high, medium, and low influences, with 9-dimensional user feature and 3-dimensional content features (see Appendix E for details). The second network is constructed to have a star-like structure, consisting of $N = 70$ users of various influence levels and $K = 1$ contents (see AppendixE Figure2b for details). Experiment result of the first network is reported in Figure 1(a), and result of the second star network is reported in Appendix E Figure 3.

**Twitter social network.** We further conduct experiments using the Twitter Social Network dataset (Leskovec & Krevl, 2014), which represents real-world social networks. The dataset contains ∼80k nodes and ∼1 million directed edges, where a directed edge $(u, v)$ means the node $u$ follows the node $v$ on Twitter. We randomly sample multiple sub-graphs from the Twitter network and construct $K$

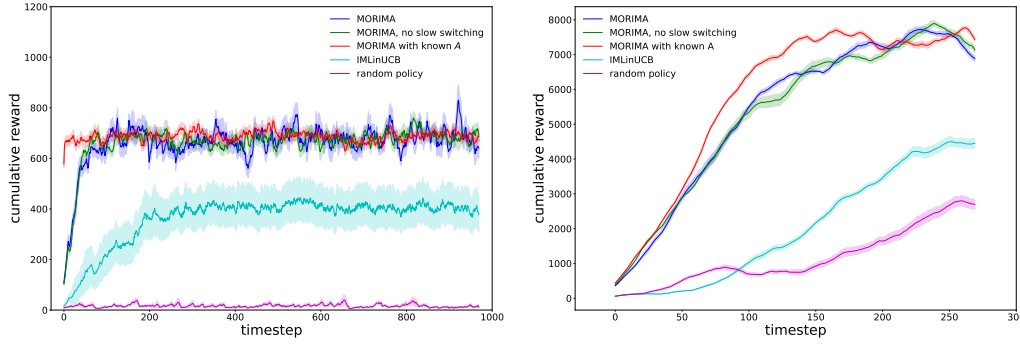

(a) Synthetic cascade network (300 nodes)    (b) Twitter network (1966 nodes)

Figure 1: **Real-time discounted sum of rewards** of IM on synthetic and Twitter networks. Averaged results with 85% CI bands are included.

networks corresponding to different topics/content types. For the sampled networks, we first pick out $n_1$ nodes with most out-degrees and then include all their $\ell$-hop neighbors, with $n_1 = 8, \ell = 5$. We construct a connectivity tensor over these networks by randomly drawing edge weights from $[0, 0.1]$ and normalizing tensor row sums to 0.9. Next we apply non-negative Tucker decomposition (Kossaifi et al., 2019) to extract the Tensor core model $\mathcal{T}^*$, user features (of dimension $d_1 = 10$) and content features (of dimension $d_2 = 2$). Thus, we have generated a large-scale topic-aware Twitter diffusion network with $K = 3$ content types, $N = 1966$ nodes, and 38023 edges for dynamic influence maximization.

**Implementation and Baselines.** Exactly solving for the optimal policy, even if the network is fully known, requires solving a combinatorially hard planning problem and is intractable. In our experiment, we adopt the two-step lookahead approximate dynamic programming scheme (Powell, 2007) as the planning oracle for Algorithm 2 of MORIMA. In the implementation of Algorithm 1, we set $\gamma = 0.9, \lambda = 1$ for synthetic networks and $\lambda = 0.01$ for the Twitter network.

For comparison with MORIMA, we also test the following baselines: (i) the naive random policy that uniformly selects a user-content pair to activate; (2) the IMLinUCB (Wen et al., 2017) which is combinatorial linear bandits baseline that was originally designed for non-adaptive online IM. To make a fair comparison with the two-step lookahead oracle, we run IMLinUCB every 2 timesteps and play the 2 selected actions spontaneously; (3) MORIMA without slow switching, where we force the Q-function to be updated at each time step; (4) MORIMA with known $\mathbf{A}^k$s - purely planning with a fully know model - it would be a performance upper bound of the reinforcement learning algorithm.

**Results and analysis.** We report the averaged discounted cumulative rewards and its empirical confidence region of our experiments in Figure 1, where each test is repeated on synthetic networks for 20 times and Twitter network for 5 times. In Figure 1 (a), we observe that the discounted sum of reward of MORIMA reaches the same level of the performance upper bound with true $\mathbf{A}$ in less than 100 rounds on synthetic network, showing that our algorithm can quickly explore the unknown network and learn to make optimal decisions. In Figure 1 (b), we observe that MORIMA can still match the performance of the planning oracle in large real social network with thousands of users. Across all experiments, our reinforcement learning-based MORIMA significantly outperforms IMLinUCB because it can adaptively make decisions based on current state while IMLinUCB does not take state into consideration. Further, slow switching did not hurt the performance of MORIMA while greatly reducing the computation complexity from $O(T)$ parameter updates to $\mathcal{O}(d \log T)$.

# 6    CONCLUSION

In this paper, we study the problem of content-aware online adaptive influence maximization and formulate the problem as an infinite-horizon discount MDP. We propose MORIMA, a model-based reinforcement learning algorithm that learns optimal policy using only node-level feedback under the IC model. We provide a $\widetilde{\mathcal{O}}(d\sqrt{T}/\Delta + \sqrt{dNKT})$ regret bound for our algorithm, which is the first sublinear regret bound of online adaptive influence maximization problem. We empirically validated the effectiveness of our algorithm on synthetic and real-world social networks.

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

# A LIMITATIONS AND BROADER IMPACT

As a common limitation for all previous work on influence maximization, it is computationally infeasible to find the exact optimal policy, especially when applied to real-world networks with millions of nodes. Therefore, the scalability of our proposed algorithm comes at the cost of using approximations. Given a certain computational budget constraint, the optimality of the policy needs to be traded-off towards affordable computational and storage complexity. In our experiments, we use Monte-Carlo methods, parallel computation and randomized tree search (dynamic programming) methods for approximating the optimal policies. While the real-world social network graph in our experiment is in the same scale as the graphs used in previous online IM studies, scaling up to larger network is an important future work of ours.

In the paper, we propose a model-based RL algorithm to learn the optimal policy for online adaptive influence maximization problems, which can be applied to advertisements for promoting beneficial ideas, new knowledge, and innovative products across social networks. However, such algorithms might be exploited to propagate fake news or rumors through the social networks. Addressing the ethical concerns also needs to be considered in future work.

# B A SHARPER BOUND FOR THE CONFIDENCE REGION

## B.1 MAIN LEMMA

**Lemma B.1** (Confidence Region). *Let Assumptions 1-4 hold. With probability at least $1 - \delta$, we have for all $t \geq 1$,*

$$\|\widehat{\mathcal{T}}_t - \mathcal{T}^*\|_{\Sigma_{t-1}} \leq \beta_t,$$

*where*

$$\beta_t = \Big(\frac{24}{\Delta}\sqrt{C/(NK) \cdot d \cdot \log(1 + NKL^2t/(d\lambda))} + 4\Big)\log(8N^2K^2t^2/\delta) + \sqrt{\lambda}\|\mathcal{T}^*\|_2.$$

*and $L = \sup\|\phi_{i,k}^t\|_2$, $d = d_1^2 d_2$.*

Before the proof of Lemma B.1, we introduce two lemmas below:

**Lemma B.2** (High probability bounds for the number of active user-content pairs). *Let Assumptions 1-4 hold. For any possibly non-stationary policy $\pi$, with probability at least $1 - \delta$, we have for all $t \geq 1$,*

$$\|s_t\|_1 \leq \frac{2}{\Delta}\Big(\frac{2}{\Delta}\log\frac{2t^2}{\delta} + 1\Big).$$

**Lemma B.3** (Bernstein-type self-normalized bound, batched version (Zhou et al., 2021a)). *Let $\{\mathcal{F}_t\}_{t=1}^{\infty}$ be a filtration, $\{x_t^i, y_t^i\}_{t \geq 1, 1 \leq i \leq m}$ be a stochastic process such that $x_t^i \in \mathbb{R}^d$ is $\mathcal{F}_t$-measurable and $y_t^i \in \mathbb{R}$ is $\mathcal{F}_{t+1}$-measurable. Assume that conditioned on $\mathcal{F}_t$, $\{y_t^1, \cdots, y_t^m\}$ are independent, and*

$$|y_t^i| \leq R, \quad \mathbb{E}[y_t^i|\mathcal{F}_t] = \langle \mathcal{T}^*, x_t^i\rangle, \quad \text{var}[y_t^i|\mathcal{F}_t] \leq \sigma^2, \quad \|x_t^i\|_2 \leq L,$$

*then with probability at least $1 - \delta$, the following holds simultaneously for all $t \geq 1$:*

$$\|\widehat{\mathcal{T}}_t - \mathcal{T}^*\|_{\Sigma_{t-1}} \leq \beta_t, \quad \Big\|\sum_{i=1}^m\sum_{\tau=1}^{t-1}x_\tau^i(y_\tau^i - \langle\mathcal{T}^*, x_\tau^i\rangle)\Big\|_{\Sigma_{t-1}^{-1}} \leq \beta_t - \sqrt{\lambda}\|\mathcal{T}^*\|_2.$$

*where $\widehat{\mathcal{T}}_t = \Sigma_{t-1}^{-1}B_{t-1}$, $\Sigma_{t-1} = \lambda I + \sum_{i=1}^m\sum_{\tau=1}^{t-1}x_\tau^i(x_\tau^i)^\top$, $B_{t-1} = \sum_{i=1}^m\sum_{\tau=1}^{t-1}x_\tau^i y_\tau^i$, and*

$$\beta_t = 8\sigma\sqrt{d\log(1 + mtL^2/(d\lambda))\log(4m^2t^2/\delta)} + 4R\log(4m^2t^2/\delta) + \sqrt{\lambda}\|\mathcal{T}^*\|_2.$$

*proof of Lemma B.1.* We use Lemma B.3 for batched stochastic process $\{\phi_{i,k}^t, (s_{t+1})_{i,k}\}$. Notice that we can choose $\sigma^2$ to be the upper bound of $\text{var}[(s_{t+1})_{i,k}|s_t, a_t]$, and

$$\text{var}[(s_{t+1})_{i,k}|s_t, a_t] \leq \mathbb{E}[(s_{t+1})_{i,k}|s_t, a_t] = \sum_j \mathbf{A}_{i,j}^k(s_{ta_t})_{j,k} \leq \frac{C}{NK}\sum_j(s_{ta_t})_{j,k} \leq \frac{C}{NK}(\|s_t\|_1 + 1).$$

where we used the assumption that $\mathbf{A}^k_{i,j} \leq \frac{C}{NK}$. By Lemma B.2, we have with probability at least $1 - \delta/2$, for all $t$,

$$\|s_t\|_1 \leq \frac{2}{\Delta}(\frac{2}{\Delta} \log \frac{4t^2}{\delta} + 1).$$

Therefore, when the above inequalities hold, we have

$$\text{var}[(s_{t+1})_{i,k}|s_t, a_t] \leq \frac{C}{NK}(\frac{2}{\Delta}(\frac{2}{\Delta} \log \frac{4t^2}{\delta} + 1) + 1) \leq \frac{C}{NK}(\frac{3}{\Delta})^2 \log \frac{4t^2}{\delta}.$$

By Lemma B.3 with $m = NK$, $R = 1$, and $\sigma^2 = \frac{C}{NK}(3/\Delta)^2 \log \frac{4t^2}{\delta}$, we have

$$\beta_t = \frac{24}{\Delta}\sqrt{C/(NK) \cdot d \cdot \log(4t^2/\delta) \log(1 + NKL^2t/(d\lambda)) \log(8N^2K^2t^2/\delta)} + 4\log(8N^2K^2t^2/\delta) + \sqrt{\lambda}\|\mathcal{T}^*\|_2,$$

which is smaller than the result stated in the lemma. $\qquad\square$

## B.2 DEFERRED PROOFS IN SUBSECTION B.1

*proof of Lemma B.2.* First, we bound the expectation of $\|s_t\|_1$. By the transition, we have

$$\mathbb{E}[(s_{t+1})_{i,k}|s_t, a_t] = \sum_j \mathbf{A}^k_{i,j}(s_{ta_t})_{j,k}$$

Therefore,

$$\mathbb{E}[\|s_{t+1}\|_1|s_t, a_t] = \sum_{i,j,k} \mathbf{A}^k_{i,j}(s_{ta_t})_{j,k}$$

Recall that we have assumed that for any content $k$ and any user $j$,

$$\sum_i \mathbf{A}^k_{i,j} \leq 1 - \Delta.$$

Then we have

$$\mathbb{E}[\|s_{t+1}\|_1|s_t, a_t] \leq (1 - \Delta)\sum_{j,k}(s_{ta_t})_{j,k} = (1 - \Delta)\|s_{ta_t}\|_1 \leq (1 - \Delta)(\|s_t\|_1 + 1), \quad (9)$$

where the last inequality holds since the action alters at most one entry of the state.

Next, notice that conditioned on $(s_t, a_t)$, $\|s_{t+1}\|$ is the summation of $NK$ independent Bernoulli random variables. By Bernstein inequality, we have with probability at least $1 - \delta_t$,

$$\|s_{t+1}\|_1 - \mathbb{E}[\|s_{t+1}\|_1|s_t, a_t] \leq 2(\sqrt{\sum_{i,k} \text{var}[(s_{t+1})_{i,k}|s_t, a_t] \log \frac{1}{\delta_t}} + \log \frac{1}{\delta_t}).$$

Since the variance of a Bernoulli random variable is bounded by its expectation, we have

$$\|s_{t+1}\|_1 - \mathbb{E}[\|s_{t+1}\|_1|s_t, a_t] \leq 2(\sqrt{\mathbb{E}[\|s_{t+1}\|_1|s_t, a_t] \log \frac{1}{\delta_t}} + \log \frac{1}{\delta_t}),$$

Therefore, by Equation (9), we have

$$\|s_{t+1}\|_1 \leq (1 - \Delta)(\|s_t\|_1 + 1) + 2\sqrt{(1 - \Delta)(\|s_t\|_1 + 1) \log \frac{1}{\delta_t}} + 2\log \frac{1}{\delta_t}$$

$$\leq (1 - \Delta)(\|s_t\|_1 + 1) + \frac{\Delta}{2}(\|s_t\|_1 + 1) + \frac{2}{\Delta}(1 - \Delta) \log \frac{1}{\delta_t} + 2\log \frac{1}{\delta_t}$$

$$= (1 - \Delta/2)(\|s_t\|_1 + 1) + \frac{2}{\Delta} \log \frac{1}{\delta_t},$$

where we used $at + b/t \geq 2\sqrt{ab}$ for the last inequality.

Finally, we set $\delta_t = \frac{\delta}{2t^2}$ so that $\sum_t \delta_t \leq \delta$ and take union bound over all $t \geq 1$. By solving the recursion, we have with probability at least $1 - \delta$,

$$\|s_t\|_1 \leq \frac{2}{\Delta}(\frac{2}{\Delta} \log \frac{2t^2}{\delta} + 1)$$

for all $t \geq 1$. $\qquad\square$

*proof of Lemma B.3.* We consider a "serialized" stochastic process. Let $\mathcal{G}_{t,i} = \sigma(\mathcal{F}_t, y_t^1, \ldots, y_t^{i-1})$. When $1 \leq i \leq m$, we have $\mathcal{G}_{t,i} \subseteq \mathcal{G}_{t,i+1}$; while when $i = m+1$, we have $\mathcal{G}_{t,m+1} = \sigma(\mathcal{F}_t, y_t^1, \ldots, y_t^m) \subseteq \mathcal{G}_{t+1,1} = \mathcal{F}_{t+1}$. Then we know that

$$\mathcal{G}_{1,1} \subseteq \mathcal{G}_{1,2} \subseteq \cdots \subseteq \mathcal{G}_{1,m} \subseteq \mathcal{G}_{2,1} \subseteq \mathcal{G}_{2,2} \subseteq \cdots \subseteq \mathcal{G}_{2,m} \subseteq \cdots \subseteq \mathcal{G}_{t,1} \subseteq \mathcal{G}_{t,2} \subseteq \cdots \subseteq \mathcal{G}_{t,m} \subseteq \cdots$$

is a filtration. Clearly we have $x_t^i$ is $\mathcal{G}_{t,i}$-measurable and $y_t^i$ is $\mathcal{G}_{t,i+1}$-measurable. By the conditional independence assumption, we also have

$$y_t^i | \mathcal{G}_{t,i} =_d y_t^i | \mathcal{F}_t, y_t^1, \ldots, y_t^{i-1} =_d y_t^i | \mathcal{F}_t.$$

Therefore, by Theorem 4.1 of Zhou et al. (2021a), we have with probability at least $1 - \delta$, for all $t \geq 1$ and $i = 1, \ldots, m$,

$$\|\widehat{\mathcal{T}}_{t,i} - \mathcal{T}^*\|_{\Sigma_{t,i}} \leq \beta_{t,i},$$

and

$$\Big\| \sum_{\tau=1}^{t-1} \sum_{j=1}^{m} x_\tau^j (y_\tau^j - \langle \mathcal{T}^*, x_\tau^j \rangle) + \sum_{j=1}^{i} x_t^j (y_t^j - \langle \mathcal{T}^*, x_t^j \rangle) \Big\|_{\Sigma_{t,i}^{-1}} \leq \beta_t - \sqrt{\lambda} \|\mathcal{T}^*\|_2,$$

where

$$\widehat{\mathcal{T}}_{t,i} = \Sigma_{t,i}^{-1} B_{t,i}, \ \Sigma_{t,i} = \lambda I + \sum_{\tau=1}^{t-1} \sum_{j=1}^{m} x_\tau^j (x_\tau^j)^\top + \sum_{j=1}^{i} x_t^j (x_t^j)^\top, B_{t,i} = \sum_{\tau=1}^{t-1} \sum_{j=1}^{m} x_\tau^j y_\tau^j + \sum_{j=1}^{i} x_t^j y_t^j,$$

and

$$\beta_{t,i} = 8\sigma \sqrt{d \log(1 + t_i L^2/(d\lambda)) \log(4t_i^2/\delta)} + 4R \log(4t_i^2/\delta) + \sqrt{\lambda} \|\mathcal{T}^*\|_2, \quad t_i = m(t-1) + i.$$

Then the result of Lemma B.3 follows by setting $i = m$. $\qquad\square$

## C   PROOF OF THEOREM 2

**Additional Notation.**

Let $t_1 = 1$, and for $s \geq 1$, the next switching time $t_{s+1}$ is recursively defined as

$$t_{s+1} = \min\{t | \det(\Sigma_{t-1}) > 2 \det(\Sigma_{t_s-1})\}.$$

Denote the set of switching times by $W = \{t_1, t_2, \ldots, t_M\}$ where $M$ is the total number of switches. We have $1 = t_1 < t_2 < \cdots < t_M \leq T < t_{M+1}$. We slightly abuse the notation and use $t_s$ to denote the last switch up to time $t$, i.e., $t_s \leq t < t_{s+1}$. Then by slow switching we mean $Q_t = Q^*_{\widehat{\mathcal{T}}_{t_s}, \widetilde{r} + b_{t_s}}$.

Recall the definition of the regrets

$$\text{Regret}(T) = \sum_{t=1}^{T} (V^*(s_t) - V_t^\pi(s_t)), \quad \widetilde{\text{Regret}}(T) = \sum_{t=1}^{T} (\widetilde{V}^*(s_t) - \widetilde{V}_t^\pi(s_t)),$$

where $V, \text{Regret}$ are defined with the original untruncated model and $\widetilde{V}, \widetilde{\text{Regret}}$ are defined with the truncated-reward model.

**Key Lemmas.** Before the proof of Theorem 2, we introduce several key lemmas.

**Lemma C.1** (optimism)**.** *Let Assumptions 1-4 hold. Set the bonus term to be*

$$b_t(s,a) = \frac{2\Lambda\gamma}{1-\gamma} \sum_{i=1}^{N} \sum_{k=1}^{K} (1 \wedge \beta_t \cdot \|\phi_{i,k}(s,a)\|_{\Sigma_{t-1}^{-1}}).$$

*Then with probability at least $1 - \delta$, we have the optimistic condition $\widetilde{Q}^*(s,a) \leq Q_t(s,a)$ holds for all $t \geq 1$.*

*Furthermore, we have for any $V(s)$ such that $0 \leq V(s) \leq \Lambda/(1-\gamma)$,*

$$\gamma |\mathbb{E}_{s' \sim \mathbb{P}(s'|s,a)} V(s') - \mathbb{E}_{s' \sim \mathbb{P}_{\widehat{\mathcal{T}}_t}(s'|s,a)} V(s')| \leq b_t(s,a).$$

**Lemma C.2** (surrogate regret). *Let Assumptions 1-4 hold. Assume that $T \log(1/\gamma) \geq \log(2NKT)$. For any policy $\pi$, we have the following connection of the regrets of the two MDPs.*

$$\text{Regret}(T) \leq \widetilde{\text{Regret}}(T) + \frac{1}{1-\gamma}.$$

**Lemma C.3** (regret decomposition (Zhou et al., 2021b)). *Let Assumptions 1-4 hold. Assume at each time step t, the results of Lemma C.1 holds. Then with probability at least $1 - \delta$, we have the following regret decomposition*

$$\widetilde{\text{Regret}}(T) \leq \frac{1}{1-\gamma}\Big[2\sum_{t=1}^{T} b_{t_s}(s_t, a_t) + \frac{2\gamma\Lambda}{1-\gamma}\sqrt{T\log\frac{1}{\delta}} + \gamma\Big(2\Lambda/(1-\gamma) + E_T\Big)\Big].$$

*where $E_T$ is the switching error*

$$E_T = \sum_{t=1}^{T} V_t(s_{t+1}) - V_{t+1}(s_{t+1}).$$

**Lemma C.4** (bounding the number of switches). *Let Assumptions 1-4 hold. The total number of the switches $M$ incurred by Algorithm 1 is bounded as*

$$M < \frac{1}{\log 2} d \log\Big(\frac{d + NKTL^2/\lambda}{d}\Big) + 1,$$

*where $L = \sup\|\phi_{i,k}^t\|_2$.*

Next we state the proof of Theorem 2.

*proof of Theorem 2.* Combing Lemma C.1, Lemma C.2, and Lemma C.3, we have with probability at least $1 - 2\delta$, when $T \log(1/\gamma) \geq \log(2NKT)$,

$$\text{Regret}(T) \leq \frac{1}{1-\gamma}\Big[2\sum_{t=1}^{T} b_{t_s}(s_t, a_t) + \frac{2\gamma\Lambda}{1-\gamma}\sqrt{T\log\frac{1}{\delta}} + \gamma\Big(2\Lambda/(1-\gamma) + E_T\Big)\Big] + \frac{1}{1-\gamma}.$$

Next we provide an upper bound for $\sum_{t=1}^{T} b_{t_s}(s_t, a_t)$. By Lemma C.1 we know that

$$b_{t_s}(s_t, a_t) \leq \frac{2\gamma\Lambda}{1-\gamma}\beta_T \sum_{i=1}^{N}\sum_{k=1}^{K}(1 \wedge \|\phi_{i,k}(s_t, a_t)\|_{\Sigma_{t_s-1}^{-1}}).$$

For any $(i, k)$, define $\Sigma_{t,i,k} = \Sigma_{t-1} + \sum_{j=1}^{i-1}\sum_{l=1}^{K}\phi_{j,l}^t(\phi_{j,l}^t)^\top + \sum_{l=1}^{k}\phi_{i,l}^t(\phi_{i,l}^t)^\top$. By the definition that $t_{s+1} = \min\{t| \det(\Sigma_{t-1}) > 2\det(\Sigma_{t_s-1})\}$, we have $\det(\Sigma_{t_{s+1}-2}) \leq 2\det(\Sigma_{t_s-1})$. Therefore, when $t_s \leq t < t_{s+1} - 1$, we have

$$\det(\Sigma_{t,i,k}) \leq \det(\Sigma_t) \leq \det(\Sigma_{t_{s+1}-2}) \leq 2\det(\Sigma_{t_s-1}).$$

By Lemma D.4, this implies

$$\|\phi_{i,k}(s_t, a_t)\|_{\Sigma_{t_s-1}^{-1}}^2 \leq 2\|\phi_{i,k}(s_t, a_t)\|_{\Sigma_{t,i,k-1}^{-1}}^2.$$

Then we have

$$\sum_{t=1}^{T}\sum_{i=1}^{N}\sum_{k=1}^{K}(1 \wedge \|\phi_{i,k}(s_t, a_t)\|_{\Sigma_{t_s-1}^{-1}})$$

$$= \sum_{t+1 \in W}\sum_{i=1}^{N}\sum_{k=1}^{K}(1 \wedge \|\phi_{i,k}(s_t, a_t)\|_{\Sigma_{t_s-1}^{-1}}) + \sum_{t+1 \notin W}\sum_{i=1}^{N}\sum_{k=1}^{K}(1 \wedge \|\phi_{i,k}(s_t, a_t)\|_{\Sigma_{t_s-1}^{-1}})$$

$$\leq MNK + \sqrt{NKT \sum_{t+1 \notin W}\sum_{i=1}^{N}\sum_{k=1}^{K}(1 \wedge \|\phi_{i,k}(s_t, a_t)\|_{\Sigma_{t_s-1}^{-1}}^2)}$$

$$\leq MNK + \sqrt{2NKT \sum_{t+1 \notin W}\sum_{i=1}^{N}\sum_{k=1}^{K}(1 \wedge \|\phi_{i,k}(s_t, a_t)\|_{\Sigma_{t,i,k-1}^{-1}}^2)}$$

$$\leq MNK + \sqrt{2NKT \sum_{t=1}^{T}\sum_{i=1}^{N}\sum_{k=1}^{K}(1 \wedge \|\phi_{i,k}(s_t, a_t)\|_{\Sigma_{t,i,k-1}^{-1}}^2)}$$

$$\leq MNK + \sqrt{2NKT \cdot 2d \log \frac{d\lambda + NKTL^2}{d\lambda}},$$

where the last inequality follows from Lemma D.3. This implies

$$\sum_{t=1}^{T} b_{t_s}(s_t, a_t) \leq \frac{2\gamma\Lambda}{1-\gamma}\beta_T\Big(MNK + \sqrt{4NKdT \log \frac{d\lambda + NKTL^2}{d\lambda}}\Big).$$

Next we bound the switching error $E_T$. Since there are in total $M$ switches, we know that there are at most $M$ non-zero terms in the summation of $E_T$. Then we have

$$E_T = \sum_{t=1}^{T}(V_t(s_{t+1}) - V_{t+1}(s_{t+1})) \leq \frac{\Lambda}{1-\gamma}M.$$

Plugging the result of Lemma C.4, we have $E_T = \widetilde{\mathcal{O}}(\frac{\Lambda}{1-\gamma}d)$.

Therefore, we have the final regret upper bound when $T \log(1/\gamma) \geq \log(2NKT)$:

$$\text{Regret}(T) \leq \widetilde{\mathcal{O}}\Big(\frac{1}{\Delta^2(1-\gamma)^2} \cdot \big(d\sqrt{C}/\Delta + \sqrt{dNK}\big) \cdot \sqrt{T}\Big) + \text{polylog}(T)\text{-terms},$$

where $d = \dim(\mathcal{T}^*) = d_1^2 d_2$. When $T \log(1/\gamma) \leq \log(2NKT)$, the above inequality trivially holds. $\qquad\square$

## C.1 DEFERRED PROOFS

*proof of Lemma C.1.* For simplicity, define $\widehat{\mathbb{P}} = \mathbb{P}_{\widehat{\mathcal{T}}_t}$, which is the estimated transition distribution obtained using $\widehat{\mathcal{T}}_t$. Notice that $\mathbb{P}_{i,k}(\cdot|s, a)$ is a Bernoulli distribution with success probability $\langle \mathcal{T}^*, \phi_{i,k}(s, a)\rangle$, while $\mathbb{P}_{i,k}(\cdot|s, a)$ is a Bernoulli distribution with success probability $\langle \widehat{\mathcal{T}}_t, \phi_{i,k}(s, a)\rangle$. Therefore, we have

$$\|\mathbb{P}_{i,k}(\cdot|s, a) - \widehat{\mathbb{P}}_{i,k}(\cdot|s, a)\|_1 \leq 2(1 \wedge |\langle \mathcal{T}^* - \widehat{\mathcal{T}}_t, \phi_{i,k}(s, a)\rangle|).$$

By Lemma B.1, with probability at least $1 - \delta$, we have $\|\mathcal{T}^* - \widehat{\mathcal{T}}_t\|_{\Sigma_{t-1}} \leq \beta_t$ for all $t \geq 1$. Then by Cauchy Inequality, the above term can be further bounded as

$$\|\mathbb{P}_{i,k}(\cdot|s, a) - \widehat{\mathbb{P}}_{i,k}(\cdot|s, a)\|_1 \leq 2(1 \wedge \beta_t \cdot \|\phi_{i,k}(s, a)\|_{\Sigma_{t-1}^{-1}}).$$

By Lemma D.2, we have

$$\|\mathbb{P}(\cdot|s, a) - \widehat{\mathbb{P}}(\cdot|s, a)\|_1 \leq \sum_{i,k}\|\mathbb{P}_{i,k}(\cdot|s, a) - \widehat{\mathbb{P}}_{i,k}(\cdot|s, a)\|_1 \leq 2\sum_{i,k}(1 \wedge \beta_t \cdot \|\phi_{i,k}(s, a)\|_{\Sigma_{t-1}^{-1}}).$$

Then by Lemma D.1, we know the desired result holds. $\qquad\square$

*proof of Lemma C.2.* Define $\pi^*$ as the optimal policy under the original model. Then we have

$$
\begin{aligned}
V^*(s_t) - \widetilde{V}^*(s_t) &= V^*(s_t) - \widetilde{V}^{\pi^*}(s_t) + \widetilde{V}^{\pi^*}(s_t) - \widetilde{V}^*(s_t) \\
&\leq V^*(s_t) - \widetilde{V}^{\pi^*}(s_t) \\
&= \mathbb{E}^{\pi^*}\Big[\sum_{i=1}^{\infty} \gamma^{i-1} r(s_i, a_i)\Big|s_1 = s_t\Big] - \mathbb{E}^{\pi^*}\Big[\sum_{i=1}^{\infty} \gamma^{i-1}\widetilde{r}(s_i, a_i)\Big|s_1 = s_t\Big],
\end{aligned}
$$

where the inequality holds because $\widetilde{V}^*(s_t)$ is the optimal V-function with respect to the truncated-reward model.

Notice that by Lemma B.2 with $\delta_0 = 1/(2NKT)$, we know that for policy $\pi^*$, with probability at least $1 - \delta_0$, we have $\|s_i\|_1 \leq \Lambda = \frac{6}{\Delta^2}\log(4NKT^3)$ for all $1 \leq i \leq T$. Therefore, with probability at least $1 - \delta_0$, $r(s_i, a_i) = \widetilde{r}(s_i, a_i)$ for all $1 \leq i \leq T$. Then

$$
\begin{aligned}
V^*(s_t) - \widetilde{V}^*(s_t) &= \mathbb{E}^{\pi^*}\Big[\sum_{i=1}^{\infty} \gamma^{i-1}(r(s_i, a_i) - \widetilde{r}(s_i, a_i))\Big|s_1 = s_t\Big] \\
&= \mathbb{E}^{\pi^*}\Big[\sum_{i=1}^{T} \gamma^{i-1}(r(s_i, a_i) - \widetilde{r}(s_i, a_i))\Big|s_1 = s_t\Big] + \mathbb{E}^{\pi^*}\Big[\sum_{i=T+1}^{\infty} \gamma^{i-1}(r(s_i, a_i) - \widetilde{r}(s_i, a_i))\Big|s_1 = s_t\Big] \\
&\leq (1 - \delta_0) \cdot 0 + \delta_0 \sum_{i=1}^{T} \gamma^{i-1} NK + NK\frac{\gamma^T}{1 - \gamma} \\
&\leq \frac{1}{1 - \gamma}\delta_0 NK + \frac{1}{1 - \gamma}\delta_0 NK = \frac{1}{T(1 - \gamma)},
\end{aligned}
$$

where the last inequality holds when $T\log(1/\gamma) \geq \log(2NKT)$.

On the other hand, we have

$$
V_t^{\pi}(s_t) - \widetilde{V}_t^{\pi}(s_t) = \mathbb{E}^{\pi}\Big[\sum_{i=0}^{\infty} \gamma^{t+i}(r(s_{t+i}, a_{t+i}) - \widetilde{r}(s_{t+i}, a_{t+i}))\Big|s_1, \cdots, s_t\Big] \geq 0.
$$

Therefore,

$$
\text{Regret}(T) \leq \widetilde{\text{Regret}}(T) + \sum_{t=1}^{T}[(V^*(s_t) - \widetilde{V}^*(s_t)) - (V_t^{\pi}(s_t) - \widetilde{V}_t^{\pi}(s_t))] \leq \widetilde{\text{Regret}}(T) + \frac{1}{1 - \gamma}.
$$

$\square$

*proof of Lemma C.3.* Define $V_t(s) = \max_a Q_t(s, a)$. By the assumption that $\widetilde{Q}^*(s, a) \leq Q_t(s, a)$, we have $\widetilde{V}^*(s) \leq V_t(s)$. Then

$$
\widetilde{\Delta}_t = \widetilde{V}^*(s_t) - \widetilde{V}_t^{\pi}(s_t) \leq V_t(s_t) - \widetilde{V}_t^{\pi}(s_t) = Q_t(s_t, a_t) - \widetilde{V}_t^{\pi}(s_t),
$$

where the last equality holds since $a_t \in \text{argmax}_a Q_t(s_t, a)$, i.e., we take the action $a_t$ greedily according to $Q_t$.

The optimal truncated Q-function $Q^*_{\widehat{\mathcal{T}}_{t_s}, \widetilde{r}+b_{t_s}}(s, a)$ (Algorithm 2) satisfies the following truncated Bellman equation:

$$
Q^*_{\widehat{\mathcal{T}}_{t_s}, \widetilde{r}+b_{t_s}}(s, a) = \min\{\Lambda/(1 - \gamma), \widetilde{r}(s, a) + b_{t_s}(s, a) + \gamma\mathbb{E}_{s' \sim \mathbb{P}_{\widehat{\mathcal{T}}_{t_s}}(\cdot|s,a)} \max_{a'} Q^*_{\widehat{\mathcal{T}}_{t_s}, \widetilde{r}+b_{t_s}}(s', a')\}.
\tag{10}
$$

Then we have

$$
\begin{aligned}
\widetilde{\Delta}_t &= \min\left\{\widetilde{r}(s_t, a_t) + b_{t_s}(s_t, a_t) + \gamma \mathbb{E}_{s' \sim \mathbb{P}_{\widehat{T}_{t_s}}(\cdot|s_t, a_t)} V_t(s'), \Lambda/(1-\gamma)\right\} \\
&\quad - \left(\widetilde{r}(s_t, a_t) + \gamma \mathbb{E}_{s' \sim \mathbb{P}(\cdot|s_t, a_t)} \widetilde{V}_{t+1}^{\pi}(s')\right) \\
&\leq \left(\widetilde{r}(s_t, a_t) + b_{t_s}(s_t, a_t) + \gamma \mathbb{E}_{s' \sim \mathbb{P}_{\widehat{T}_{t_s}}(\cdot|s_t, a_t)} V_t(s')\right) - \left(\widetilde{r}(s_t, a_t) + \gamma \mathbb{E}_{s' \sim \mathbb{P}(\cdot|s_t, a_t)} \widetilde{V}_{t+1}^{\pi}(s')\right) \\
&= b_{t_s}(s_t, a_t) + \gamma\left(\mathbb{E}_{s' \sim \mathbb{P}_{\widehat{T}_{t_s}}(\cdot|s_t, a_t)} V_t(s') - \mathbb{E}_{s' \sim \mathbb{P}(\cdot|s_t, a_t)} \widetilde{V}_{t+1}^{\pi}(s')\right) \\
&= b_{t_s}(s_t, a_t) + \gamma\left(\mathbb{E}_{s' \sim \mathbb{P}_{\widehat{T}_{t_s}}(\cdot|s_t, a_t)} V_t(s') - \mathbb{E}_{s' \sim \mathbb{P}(\cdot|s_t, a_t)} V_t(s')\right) \\
&\quad + \gamma\left(\mathbb{E}_{s' \sim \mathbb{P}(\cdot|s_t, a_t)} V_t(s') - \mathbb{E}_{s' \sim \mathbb{P}(\cdot|s_t, a_t)} \widetilde{V}_{t+1}^{\pi}(s')\right).
\end{aligned}
$$

For the second term, since $|V_t(s')| \leq \Lambda/(1-\gamma)$, by Lemma C.1, we have

$$
\gamma\left(\mathbb{E}_{s' \sim \mathbb{P}_{\widehat{T}_{t_s}}(\cdot|s_t, a_t)} V_t(s') - \mathbb{E}_{s' \sim \mathbb{P}(\cdot|s_t, a_t)} V_t(s')\right) \leq b_{t_s}(s_t, a_t).
$$

For the last term, we have

$$
\begin{aligned}
&\gamma\left(\mathbb{E}_{s' \sim \mathbb{P}(\cdot|s_t, a_t)} V_t(s') - \mathbb{E}_{s' \sim \mathbb{P}(\cdot|s_t, a_t)} \widetilde{V}_{t+1}^{\pi}(s')\right) \\
&= \gamma \xi_t + \gamma\left(V_t(s_{t+1}) - \widetilde{V}_{t+1}^{\pi}(s_{t+1})\right),
\end{aligned}
$$

where

$$
\xi_t = \left(\mathbb{E}_{s' \sim \mathbb{P}(\cdot|s_t, a_t)}(V_t(s') - \widetilde{V}_{t+1}^{\pi}(s')) - (V_t(s_{t+1}) - \widetilde{V}_{t+1}^{\pi}(s_{t+1}))\right).
$$

Therefore, we have

$$
\begin{aligned}
\sum_{t=1}^{T} \widetilde{\Delta}_t &\leq \sum_{t=1}^{T}\left(V_t(s_t) - \widetilde{V}_t^{\pi}(s_t)\right) \\
&\leq 2\sum_{t=1}^{T} b_{t_s}(s_t, a_t) + \gamma \sum_{t=1}^{T} \xi_t + \gamma \sum_{t=1}^{T}\left(V_t(s_{t+1}) - \widetilde{V}_{t+1}^{\pi}(s_{t+1})\right).
\end{aligned}
$$

For the last term above, we have

$$
\begin{aligned}
&\sum_{t=1}^{T}\left(V_t(s_{t+1}) - \widetilde{V}_{t+1}^{\pi}(s_{t+1})\right) \\
&= \sum_{t=1}^{T}\left(V_{t+1}(s_{t+1}) - \widetilde{V}_{t+1}^{\pi}(s_{t+1})\right) + \underbrace{\sum_{t=1}^{T}\left(V_t(s_{t+1}) - V_{t+1}(s_{t+1})\right)}_{E_T} \\
&= \sum_{t=0}^{T-1}\left(V_{t+1}(s_{t+1}) - \widetilde{V}_{t+1}^{\pi}(s_{t+1})\right) + \left[V_{T+1}(s_{T+1}) - \widetilde{V}_{T+1}^{\pi}(s_{T+1}) - V_1(s_1) + \widetilde{V}_1^{\pi}(s_1)\right] + E_T \\
&\leq \sum_{t=1}^{T}\left(V_t(s_t) - \widetilde{V}_t^{\pi}(s_t)\right) + 2\Lambda/(1-\gamma) + E_T.
\end{aligned}
$$

Notice that $\{\xi_t\}$ is a martingale difference sequence. Therefore, by Azuma-Hoeffding inequality, we have with probability at least $1 - \delta$,

$$
\sum_{t=1}^{T} \xi_t \leq \frac{2\Lambda}{1-\gamma} \sqrt{T \log \frac{1}{\delta}}.
$$

To summarize, we have

$$\widetilde{\text{Regret}}(\pi) \leq \sum_{t=1}^{T} \Big( V_t(s_t) - \widetilde{V}_t^{\pi}(s_t) \Big)$$

$$\leq 2 \sum_{t=1}^{T} b_{t_s}(s_t, a_t) + \frac{2\gamma\Lambda}{1-\gamma} \sqrt{T \log \frac{1}{\delta}} + \gamma \Big( \sum_{t=1}^{T} \Big( V_t(s_t) - \widetilde{V}_t^{\pi}(s_t) \Big) + 2\Lambda/(1-\gamma) + E_T \Big),$$

which implies

$$\widetilde{\text{Regret}}(\pi) \leq \frac{1}{1-\gamma} \Big[ 2 \sum_{t=1}^{T} b_{t_s}(s_t, a_t) + \frac{2\gamma\Lambda}{1-\gamma} \sqrt{T \log \frac{1}{\delta}} + \gamma \Big( 2\Lambda/(1-\gamma) + E_T \Big) \Big].$$

$\square$

*proof of Lemma C.4.* On the one hand, we have

$$\frac{\det(\Sigma_T)}{\det(\Sigma_0)} \geq \prod_{s=1}^{M-1} \frac{\det(\Sigma_{t_{s+1}-1})}{\det(\Sigma_{t_s-1})} > 2^{M-1}.$$

On the other hand, we also have

$$\frac{\det(\Sigma_T)}{\det(\Sigma_0)} = \det(\Sigma_0^{-1} \Sigma_T) \leq \Big( \frac{\text{Tr}(\Sigma_0^{-1} \Sigma_T)}{d} \Big)^d$$

$$= \Big( \frac{\text{Tr}(\boldsymbol{I} + \lambda^{-1} \sum_{i,k,t} \phi_{i,k}^t (\phi_{i,k}^t)^{\top})}{d} \Big)^d$$

$$\leq \Big( \frac{d + NKTL^2/\lambda}{d} \Big)^d.$$

Therefore, $M < \frac{1}{\log 2} d \log \Big( \frac{d + NKTL^2/\lambda}{d} \Big) + 1.$

$\square$

# D AUXILIARY LEMMAS

**Lemma D.1.** *Let $\mathbb{P}(s'|s, a)$ and $\widehat{\mathbb{P}}(s'|s, a)$ be two transition probabilities. Assume that $0 \leq r(s, a) \leq \Lambda$. Let $Q^*$ be optimal Q-function for MDP $\mathcal{M}_{\mathbb{P},r}$. Let $\widehat{Q}^*$ be the optimal truncated Q-function for $\mathcal{M}_{\widehat{\mathbb{P}},r+b}$, which satisfies the following equation*

$$\widehat{Q}^*(s, a) = \min\{\Lambda/(1-\gamma), r(s, a) + b(s, a) + \gamma \mathbb{E}_{s' \sim \mathbb{P}(\cdot|s, a)} \max_{a'} \widehat{Q}^*(s', a')\}.$$

*Then if*

$$b(s, a) \geq \frac{\gamma\Lambda}{1-\gamma} \|\mathbb{P}(\cdot|s, a) - \widehat{\mathbb{P}}(\cdot|s, a)\|_1,$$

*we have*

$$\widehat{Q}^*(s, a) \geq Q^*(s, a) \text{ for all } (s, a).$$

*Furthermore, we have for any $V(s)$ such that $0 \leq V(s) \leq \Lambda/(1-\gamma)$,*

$$\gamma |\mathbb{E}_{s' \sim \mathbb{P}(s'|s, a)} V(s') - \mathbb{E}_{s' \sim \widehat{\mathbb{P}}(s'|s, a)} V(s')| \leq b(s, a).$$

**Lemma D.2** (Factorization). *If $\mathbb{P}(\cdot|s, a) = \otimes_{i=1}^{N} \mathbb{P}_i(\cdot|s, a)$, $\widehat{\mathbb{P}}(\cdot|s, a) = \otimes_{i=1}^{N} \widehat{\mathbb{P}}_i(\cdot|s, a)$, then*

$$\|\mathbb{P}(\cdot|s, a) - \widehat{\mathbb{P}}(\cdot|s, a)\|_1 \leq \sum_{i=1}^{N} \|\mathbb{P}_i(\cdot|s, a) - \widehat{\mathbb{P}}_i(\cdot|s, a)\|_1.$$

**Lemma D.3** (Lemma 11 in (Abbasi-Yadkori et al., 2011)). *For any $\{x_t\}_{t=1}^{T} \subseteq \mathbb{R}^d$, let $\Sigma_t = \lambda \boldsymbol{I} + \sum_{t=1}^{T} x_t x_t^{\top}$, then we have*

$$\sum_{t=1}^{T} (1 \wedge \|x_t\|_{\Sigma_{t-1}^{-1}}^2) \leq 2d \log \frac{d\lambda + TL^2}{d\lambda},$$

*where $L = \sup \|x_t\|_2$.*

**Lemma D.4** (Lemma 12 in (Abbasi-Yadkori et al., 2011)). *Let $A, B \in \mathbb{R}^{d \times d}$ be two positive definite matrices and $A \succeq B$. Then for any $x \in \mathbb{R}^d$, we have*

$$\|x\|_A^2 \leq \|x\|_B^2 \cdot \frac{\det(A)}{\det(B)}.$$

# E  ADDITIONAL EXPERIMENTS

## E.1  SYNTHETIC CASCADE NETWORK

We conduct experiments on cascade synthetic networks as shown in Figure 2a. There are $K = 4$ identical networks are associated with different contents; each cascade network has $N = 300$ nodes with *high/medium/low* influential power, $5/20/275$ nodes respectively. More specifically, the five *high* nodes can activate theirselves and each others at the next time step with probability $0.1$; the *high* nodes can activate the *medium* nodes with probability $0.2$; the *medium* nodes can only activate the *low* nodes with probability $0.12$. Therefore, these networks would appreciate the *high* nodes as better actions with more delayed reward. In addition, the $d_2 = 3$-dim content features for $K = 4$ contents are $\{(1, 0, 0), (0, 1, 0), (0, 0, 1), (0.3, 0.3, 0.4)\}$; the $d_1 = 9$-dim user features are one-hot for each type of nodes associated with each dimension of content features. Finally, the underlying linear dynamic $\mathcal{T}^*$ can be easily computed for desired probabilities of edges. The experiment result of the cascade synthetic networks is reported in Figure1a.

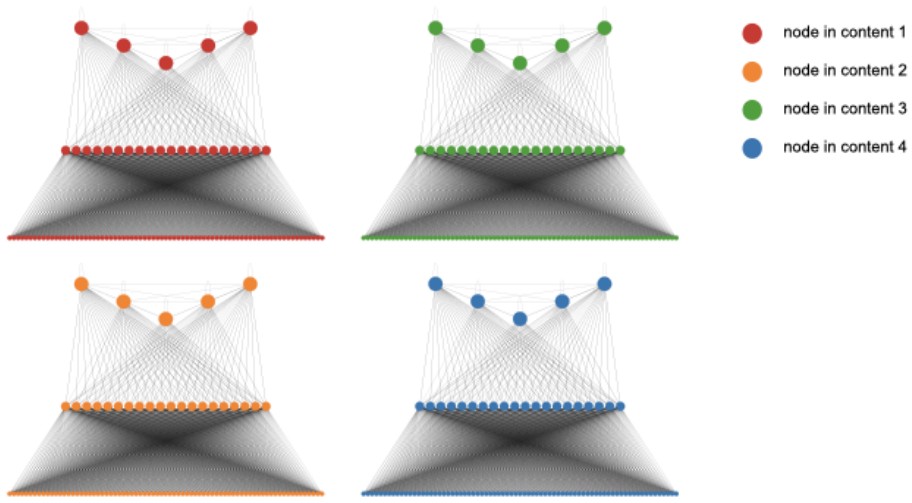

(a) Cascade network (300 nodes for each content)

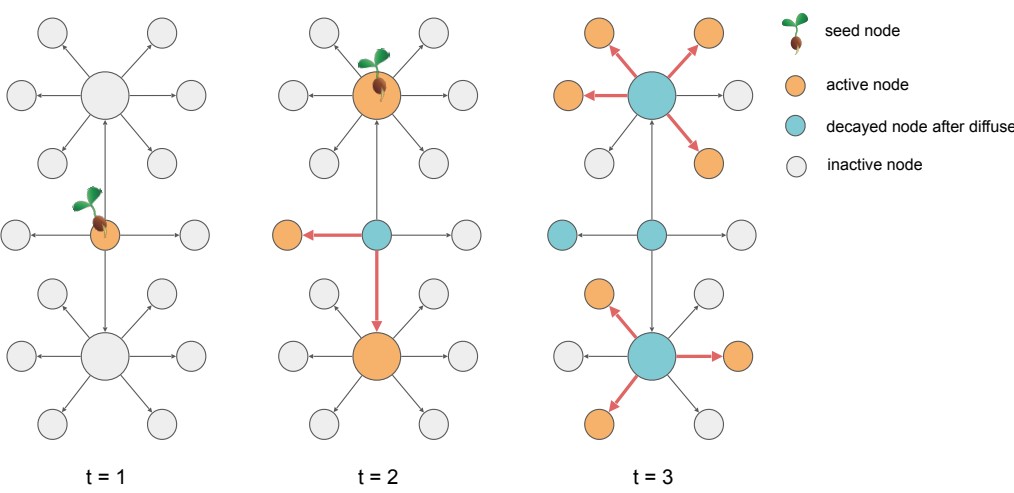

(b) Star-shape network (70 nodes)

Figure 2: **Synthetic network visualization. (a) Cascade network.** Four identical networks are associated with different content; their nodes are colored differently. The node activation could not happen across content. The cascade network has nodes with high/medium/low influential power, 5/20/275 nodes respectively. **(b) Star-shape network.** This network has three influential nodes; the center one can activate the other two influential nodes. Only edges with positive probability are visualized.

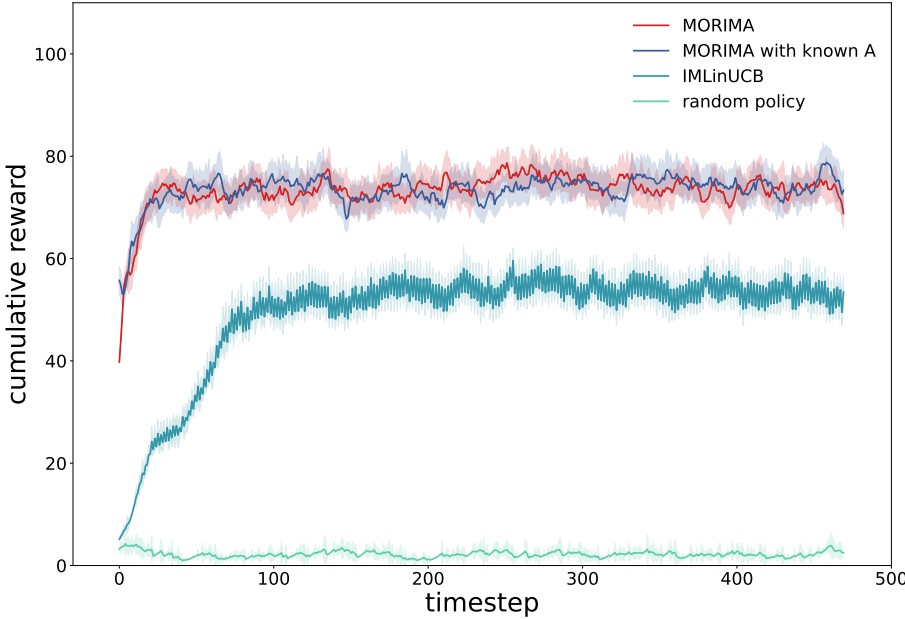

Figure 3: **Real-time discounted sum of rewards on synthetic star-shape graph (Fig.2b).** The synthetic network has static underlying dynamics $\mathcal{T}^*$ generating diffusion probabilities from each node to all other nodes. Each line is averaged from 20 trajectories; 85% CI bands are included.

## E.2 SYNTHETIC STAR-SHAPE NETWORK

We conduct additional experiments on another synthetic network as shown in Figure 2b, where a better action has more delayed reward and making decision at each time step is appreciated. This synthetic network has three influential nodes out of total $N = 70$ nodes, while the center one is the best choice and has the ability to activate the other two influential nodes. Also the central influential node has delayed but higher expected reward than expected reward by activating either of other two neighboring influential nodes. For simplicity, we set only one content is available here, $K = 1$ and $d_2 = 1$. The $d_1 = 6$ user features are one-hot vectors, indicating their neighborhood subgraph. The discount factor of reward is $\gamma = 0.9$.

In Figure 3, we compare the performance of our MORIMA to MORIMA with known $\mathbf{A}^k$ as upper bound and IMLinUCB as baseline. The details of these algorithms are the same as in section 5. MORIMA exhibits its great power to explore the unknown graph efficiently; its learning curve at the first 40 time steps overlaps with MORIMA while knowing the true dynamics $\mathbf{A}^k$. In addition, the sum of cumulative rewards of MORIMA and the upper bound stay at the same high level. Furthermore, we observe that the learning procedure of IMLinUCB takes much longer and converges to a much lower level. The classic IM setting, activating $k$ seeds at once for every $k$ step, shows its limit while adaptive decision making leads to a better result.

We ran all experiments on our internal cluster with 8 CPUs, 128G memory per task.

# F EXTENSION TO GENERALIZED LINEAR MODEL

In this section, we show how to extend our algorithm and regret bound to generalized linear models. We first re-state below the modified assumption on the transition model.

**Assumption 6** (Generalized Bernoulli Independent Cascade Model). *Let $s'$ be the next state. For each $k \in [K]$, we assume there is an underlying connectivity matrix $\mathbf{A}^k \in \mathbb{R}^{n \times n}$ such that*

$$\mathbb{P}(s'_{i,k} = 1|s) = \mu(\sum_j \mathbf{A}^k_{i,j} s_{j,k}), \tag{11}$$

*where $\mu : \mathbb{R} \to \mathbb{R}$ satisfies $\mu(0) = 0$ and $1/\kappa \leq \mu' \leq 1$ for some $\kappa \geq 1$. And we assume $s'_{i,k}$'s are independent conditioned on $s$.*

## F.1 MORIMA FOR GENERALIZED LINEAR MODEL

Next, we state the changes to our algorithm. Under this assumption, we cannot simply use ridge regression to get an empirical estimation of $\mathcal{T}^*$. Instead, we estimate the tensor model by

$$\widehat{\mathcal{T}}_t = \operatorname*{argmin}_{\mathcal{T}} L_t(\mathcal{T}) \equiv \left\| \lambda \mathcal{T} + \sum_{\tau=1}^{t-1} \sum_{k=1}^{K} \sum_{i=1}^{N} \left[ \mu(\langle \mathcal{T}, \phi^\tau_{i,k} \rangle) - (s_{\tau+1})_{i,k} \right] \phi^\tau_{i,k} \right\|_{\Sigma^{-1}_{t-1}}, \tag{12}$$

where $\phi^t_{i,k}$ and $\Sigma_t$ are defined as in Section 3.

We still perform optimistic planning with respect to the truncated-reward model, where the reward bonus term $b_t(s, a)$ is replaced with

$$b_t(s, a) = \frac{4\kappa\gamma\Lambda}{1 - \gamma} \sum_{i=1}^{N} \sum_{k=1}^{K} (1 \wedge \beta_t \cdot \|\phi_{i,k}(s, a)\|_{\Sigma^{-1}_{t-1}}), \tag{13}$$

which is the original bonus term multiplied by $2\kappa$. We adopt the same slow switching method as before.

## F.2 REGRET ANALYSIS

We have the following regret bound, which is the original bound multiplied by $\kappa$.

**Theorem 4** (Re-state Theorem 3). *Let Assumption 5, Assumptions 2-4 hold. With probability at least $1 - \delta$, Algorithm 1 satisfies the following regret upper bound:*

$$\operatorname{Regret}(T) \leq \widetilde{\mathcal{O}}\left( \frac{\kappa}{\Delta^2(1 - \gamma)^2} \cdot \left( d\sqrt{C}/\Delta + \sqrt{dNK} \right) \cdot \sqrt{T} \right) + \operatorname{polylog}(T)\text{-terms},$$

*where $d = \dim(\mathcal{T}^*) = d_1^2 d_2$.*

## F.3 PROOF SKETCH OF THEOREM 3

The proof of Theorem 3 only differs from the proof of Theorem 2 slightly. Next we examine through the proof of Theorem 2 and state the corresponding lemmas in the generalized linear model setting.

First, we have exactly the same result for the high probability bounds for the number of active user-content pairs.

**Lemma F.1** (High probability bounds for the number of active user-content pairs). *Let Assumption 5, Assumptions 2-4 hold. For any possibly non-stationary policy $\pi$, with probability at least $1 - \delta$, we have for all $t \geq 1$,*

$$\|s_t\|_1 \leq \frac{2}{\Delta}(\frac{2}{\Delta} \log \frac{2t^2}{\delta} + 1).$$

The next two lemmas justify the choice of the bonus term.

**Lemma F.2** (Confidence Region). *Let Assumption 5, Assumptions 2-4 hold. With probability at least $1 - \delta$, we have for all $t \geq 1$,*

$$L_t(\mathcal{T}^*) \leq \beta_t,$$

*where $\beta_t$ is defined as in Eqn. (6).*

**Lemma F.3** (Optimism). *Let Assumption 5, Assumptions 2-4 hold. Set the bonus term to be*

$$b_t(s,a) = \frac{4\kappa\Lambda\gamma}{1-\gamma} \sum_{i=1}^{N} \sum_{k=1}^{K} (1 \wedge \beta_t \cdot \|\phi_{i,k}(s,a)\|_{\Sigma_{t-1}^{-1}}).$$

*Then with probability at least $1 - \delta$, we have the optimistic condition $\widetilde{Q}^*(s,a) \le Q_t(s,a)$ holds for all $t \ge 1$.*

*Furthermore, we have for any $V(s)$ such that $0 \le V(s) \le \Lambda/(1-\gamma)$,*

$$\gamma|\mathbb{E}_{s'\sim\mathbb{P}(s'|s,a)}V(s') - \mathbb{E}_{s'\sim\mathbb{P}_{\widehat{\mathcal{T}}_t}(s'|s,a)}V(s')| \le b_t(s,a).$$

*proof of Theorem 3.* We can verify that Lemma C.2, Lemma C.3, and Lemma C.4 also hold for the generalized linear model. Therefore, the exact same proof of Theorem 2 applies with Lemma C.1 replaced with Lemma F.3. The result only differs by a factor of $\kappa$. □

### F.4 DEFERRED PROOFS OF LEMMAS

*proof of Lemma F.1.* Notice that we have assumed $\mu(0) = 0$ and $\mu' \le 1$. Therefore for $z > 0$, $\mu(z) = \mu(z) - \mu(0) \le z - 0 = z$. Then

$$\mathbb{E}[(s_{t+1})_{i,k}|s_t, a_t] = \mu(\sum_j \mathbf{A}_{i,j}^k(s_{ta_t})_{j,k}) \le \sum_j \mathbf{A}_{i,j}^k(s_{ta_t})_{j,k}.$$

Then the result holds by applying the same proof of Lemma B.2. □

*proof of Lemma F.2.* We have

$$L_t(\mathcal{T}^*) = \left\|\lambda\mathcal{T}^* + \sum_{\tau=1}^{t-1}\sum_{k=1}^{K}\sum_{i=1}^{N}\left[\mu(\langle\mathcal{T}^*, \phi_{i,k}^\tau\rangle) - (s_{\tau+1})_{i,k}\right]\phi_{i,k}^\tau\right\|_{\Sigma_{t-1}^{-1}}$$

$$\le \|\lambda\mathcal{T}^*\|_{\Sigma_{t-1}^{-1}} + \left\|\sum_{\tau=1}^{t-1}\sum_{k=1}^{K}\sum_{i=1}^{N}\left[\mu(\langle\mathcal{T}^*, \phi_{i,k}^\tau\rangle) - (s_{\tau+1})_{i,k}\right]\phi_{i,k}^\tau\right\|_{\Sigma_{t-1}^{-1}}$$

$$\le \|\lambda\mathcal{T}^*\|_{\lambda\mathbf{I}^{-1}} + \left\|\sum_{\tau=1}^{t-1}\sum_{k=1}^{K}\sum_{i=1}^{N}\left[\mu(\langle\mathcal{T}^*, \phi_{i,k}^\tau\rangle) - (s_{\tau+1})_{i,k}\right]\phi_{i,k}^\tau\right\|_{\Sigma_{t-1}^{-1}}$$

$$\le \sqrt{\lambda}\|\mathcal{T}^*\|_2 + (\beta_t - \sqrt{\lambda}\|\mathcal{T}^*\|_2)$$

$$= \beta_t,$$

where the last but one inequality holds by applying Lemma B.3 with the same variance upper bound as in the proof of Lemma B.1. □

*proof of Lemma F.3.* Let $\widehat{\mathbb{P}} = \mathbb{P}_{\widehat{\mathcal{T}}_t}$. By the assumption that $\mu' \le 1$, we still have

$$\|\mathbb{P}_{i,k}(\cdot|s,a) - \widehat{\mathbb{P}}_{i,k}(\cdot|s,a)\|_1 \le 2(1 \wedge |\langle\mathcal{T}^* - \widehat{\mathcal{T}}_t, \phi_{i,k}(s,a)\rangle|)$$

$$\le 2(1 \wedge \|\mathcal{T}^* - \widehat{\mathcal{T}}_t\|_{\Sigma_{t-1}} \cdot \|\phi_{i,k}(s,a)\|_{\Sigma_{t-1}^{-1}}).$$

By Lemma F.2, with probability at least $1 - \delta$, we have $L_t(\mathcal{T}^*) \le \beta_t$ and then $L_t(\widehat{\mathcal{T}}_t) \le \beta_t$ for all $t \ge 1$. Therefore, we have

$$2\beta_t \ge L_t(\widehat{\mathcal{T}}_t) + L_t(\mathcal{T}^*)$$

$$\ge \left\|\lambda\widehat{\mathcal{T}}_t + \sum_{\tau=1}^{t-1}\sum_{k=1}^{K}\sum_{i=1}^{N}\left[\mu(\langle\widehat{\mathcal{T}}_t, \phi_{i,k}^\tau\rangle) - (s_{\tau+1})_{i,k}\right]\phi_{i,k}^\tau - \lambda\mathcal{T}^* - \sum_{\tau=1}^{t-1}\sum_{k=1}^{K}\sum_{i=1}^{N}\left[\mu(\langle\mathcal{T}^*, \phi_{i,k}^\tau\rangle) - (s_{\tau+1})_{i,k}\right]\phi_{i,k}^\tau\right\|_{\Sigma_{t-1}^{-1}}$$

$$= \left\|\left[\lambda\mathbf{I} + \sum_{\tau=1}^{t-1}\sum_{k=1}^{K}\sum_{i=1}^{N}\left[\mu'(\langle\widetilde{\mathcal{T}}, \phi_{i,k}^\tau\rangle)\right]\phi_{i,k}^\tau(\phi_{i,k}^\tau)^\top\right] \cdot (\widehat{\mathcal{T}}_t - \mathcal{T}^*)\right\|_{\Sigma_{t-1}^{-1}}$$

$$\ge 1/\kappa \cdot \|\Sigma_{t-1} \cdot (\widehat{\mathcal{T}}_t - \mathcal{T}^*)\|_{\Sigma_{t-1}^{-1}}$$

$$= 1/\kappa \cdot \|\widehat{\mathcal{T}}_t - \mathcal{T}^*\|_{\Sigma_{t-1}},$$

where we use Lagrange Mean Value Theorem in the first equality and $\mu'(\cdot) \geq 1/\kappa$. Then the desired result holds by applying Lemma D.2 and Lemma D.1 in the same way as the proof of Lemma C.1. $\quad\square$

