# OpenReview forum: "Provably Efficient Reinforcement Learning for Online Adaptive Influence Maximization"
_ICLR.cc/2023/Conference — Submitted to ICLR 2023_

### Official Review · Reviewer_Asps · 2022-10-24

**Confidence:** 3
**Correctness:** 3
**Technical Novelty And Significance:** 2
**Empirical Novelty And Significance:** 2
**Recommendation:** 5

**Clarity, Quality, Novelty And Reproducibility:**

Questions:
1. Comparing to IMLinUCB, MORIMA variants seems to have low variance, so curious to understand why that's the case?
2. Among MORIMA variants, MORIMA without slow switching (in green) seems to be more stable over timesteps comparing to MORIMA (blue). Should we factor in stability over timesteps in choosing these variants? If so, how?
3. In Figure (b), "MORIMA with known A" seems to underperform the other two MORIMA variants when timesteps reaches 200+. Is that expected or why is that?

**Strength And Weaknesses:**

Strengths:
1. The paper proposes a network diffusion process to model (user, content) feature dependent network propagation and formulate the problem as an infinite-horizon discounted MDP, with the goal to select seed users and customize contents to maximize network influence.
2. it proves a sub-linear regret bound for the proposed algorithms, and validate their results empirically on both synthetic and twitter social network datasets.


Weakness:
The paper only compares to random policy and IMLinUCB, lack of strong comprehensive baseline comparison and empirical evaluation.



**Summary Of The Paper:**

The paper proposes a network diffusion process to model (user, content) feature dependent network propagation and formulate the problem as an infinite-horizon discounted MDP, with the goal to select seed users and customize contents to maximize network influence. Then it proves a sub-linear regret bound for the proposed algorithms, and validate their results empirically on both synthetic and twitter social network datasets.

**Summary Of The Review:**

The paper proposes a network diffusion process to model (user, content) feature dependent network propagation and formulate the problem as an infinite-horizon discounted MDP, with the goal to select seed users and customize contents to maximize network influence. Then it proves a sub-linear regret bound for the proposed algorithms, and validate their results empirically on both synthetic and twitter social network datasets.

In the experimentation section, the paper only compares to random policy and IMLinUCB, lack of strong comprehensive baseline comparison and empirical evaluation. In addition to that, there are some unclarity in the experimentation results, shared in below:
1. Comparing to IMLinUCB, MORIMA variants seems to have low variance, so curious to understand why that's the case?
2. Among MORIMA variants, MORIMA without slow switching (in green) seems to be more stable over timesteps comparing to MORIMA (blue). Should we factor in stability over timesteps in choosing these variants? If so, how?
3. In Figure (b), "MORIMA with known A" seems to underperform the other two MORIMA variants when timesteps reaches 200+. Is that expected or why is that?

---

> ### Author Response · Authors · 2022-11-18
> **Thank you for your valuable feedback**
>
> **Q1**:  lack of strong comprehensive baseline comparison and empirical evaluation.
>
> **A1**: We would like to emphasize that we are the first to properly define and study the online adaptive influence maximization (IM) problem. Therefore, there is no previous work for us to directly compare with. IMLinUCB is one of the strong baseline methods for the **static** IM problem. Our experiments show that the static IMLinUCB algorithm performs worse than our dynamic algorithm. Furthermore, our theoretical results indicate the optimality of our proposed algorithm.
>
>
> **Q2**: Compared to IMLinUCB, MORIMA variants seem to have low variance, so curious to understand why that's the case.
>
> **A2**: We conjecture that MORIMA enters the optimal state faster than IMLinUCB, where MORIMA saturates and the rewards only change in a relatively small magnitude. Instead, IMLinUCB struggles with finding the optimal state and the optimal policy, so it has a larger variance compared to MORIMA. This phenomenon is persistent among different random seeds.
>
>
> **Q3**: Among MORIMA variants, MORIMA without slow switching (in green) seems to be more stable over timesteps compared to MORIMA (blue). Should we factor in stability over timesteps in choosing these variants? If so, how?
>
> **A3**: For theoretical consideration, MORIMA with slow switching can achieve the optimal regret, while MORIMA without slow switching cannot. As for the implementation, slow switching saves computation, as we do not need to update the policy during the data collection phase. Furthermore, for the Twitter graph (fig.1 (b)) we do not observe this kind of stability difference between the two variants. Therefore, we favor MORIMA over MORIMA without slow switching.
>
> **Q4**: In Figure (b), "MORIMA with known A" seems to underperform the other two MORIMA variants when timesteps reach 200+. Is that expected or why is that?
>
> **A4**: This is contingent and caused by the noise in the simulation. From the plot, we see that after 200 timesteps, the oracle (MORIMA with known A) has already converged to the optimal state distribution but the cumulative reward still fluctuates between 7100~7900. This suggests that there is irreducible noise inherent in the environment. The other two MORIMA variants fluctuate in the reasonable region. It is normal that some variant supersede others. Therefore, from this contingent phenomenon, we cannot conclude that when timesteps reach 200+ "MORIMA with known A" underperforms the other two MORIMA variants.
>
> ----
>
> We notice that you only give 2 points on the Technical Novelty And Significance. We would like to emphasize that our paper is the first to formulate the Influence maximization problem into an  RL framework and provide sharp regret bounds for our proposed algorithm. Therefore, we’d appreciate it if you consider re-evaluating our theoretical parts.
>
> Given the above reply, could you kindly consider increasing the overall score?

---

### Official Review · Reviewer_vMK7 · 2022-10-25

**Confidence:** 4
**Correctness:** 3
**Technical Novelty And Significance:** 3
**Empirical Novelty And Significance:** 2
**Recommendation:** 3

**Clarity, Quality, Novelty And Reproducibility:**

The main question about the proposed algorithm is its scalability on huge graphs with millions of users. It is not obvious how the proposed algorithm can be applied in practice. For this purpose, experiments should be conducted on larger graphs in order to validate the algorithm's ability to maximize the influence of specific contents. Even the `real-world` networks are pretty small and are technically created by sampling multiple dense sub-graphs from the Twitter network.

Also, some points should be discussed further by the authors.
- For instance, is not clear how the states are represented in the Q-learning algorithm.
- How the agent is able to efficiently explore the environment? Is it achieved by using the bonus term?
- What happens in the case where N is extremely large? Is it still possible to compute the covariance matrix $\Sigma_{t-1}$?
- Matrix $A^k$ is not well defined. The sum of the elements of each row should be equal to one.
- The idea that a user has more than one chance to influence their neighbors and that of cumulative reward is not new.



**Strength And Weaknesses:**

**Strengths**
- The examination of the online influence maximization problem under the adaptive setting is quite interesting, as resembles what happens in the real-world.
- The adoption of a model-based reinforcement learning algorithm for selecting seed users adaptively.
- A regret bound is provided for the proposed algorithm.

**Weaknesses**
- In general, the paper is well written, nevertheless, some of its parts need to be clarified.
- The exploration efficiency of the proposed reinforcement learning algorithm is not clear.
- The applicability of the MORIMA algorithm in real-world networks with millions of users seems to be not possible.
- Experiments have been conducted only on small synthetic and real-world networks that are

**Summary Of The Paper:**

The adaptive version of the online adaptive topic-aware influence maximization problem is considered in this work, where its primary objective is the spreading of specific content in social networks. The problem is formulated as an infinite-horizon discounted MDP, and a model-based reinforcement algorithmic scheme, called MORIMA, has been proposed that learns a policy adaptively. The authors provide a regret analysis for the proposed algorithm. Experiments have been conducted on small synthetic and real-world networks, while the proposed algorithm has been compared with the IMLinUCB algorithm.


**Summary Of The Review:**

The authors should address the limitations of the MORIMA algorithm in a clear way. As aforementioned, the main question of this work is its applicability in real-world networks with a large number of users. The potential negative social impact of this work should be discussed further. How can this work have a positive impact on our society and what is its main objective?

---

> ### Author Response · Authors · 2022-11-18
> **Thank you for your valuable review**
>
> We sincerely thank the reviewer for the suggestions on the limitations and broader impact section. We have revised the paper to include the following:
>
> **Limitations and Broader Impact:**
>
> As a common limitation for all previous work on influence maximization, it is computationally infeasible to find the exact optimal policy, especially when applied to real-world networks with millions of nodes. Therefore, the scalability of our proposed algorithm comes at the cost of using approximations. Given a certain computational budget constraint, the optimality of the policy needs to be traded-off towards affordable computational and storage complexity. In our experiments, we use Monte-Carlo methods, parallel computation and randomized tree search (dynamic programming) methods for approximating the optimal policies. While the real-world social network graph in our experiment is in the same scale as the graphs used in previous online IM studies, scaling up to larger network is an important future work of ours.
>
> In the paper, we propose a model-based RL algorithm to learn the optimal policy for online adaptive influence maximization problems, which can be applied to advertisements for promoting beneficial ideas, new knowledge, and innovative products across social networks. However, such algorithms might be exploited to propagate fake news or rumors through the social networks. Addressing the ethical concerns also needs to be considered in future work.

---

> ### Author Response · Authors · 2022-11-18
> **Thank you for your valuable review (2)**
>
>
> **Q1**: The main question about the proposed algorithm is its scalability on huge graphs with millions of users. It is not obvious how the proposed algorithm can be applied in practice. For this purpose, experiments should be conducted on larger graphs in order to validate the algorithm's ability to maximize the influence of specific contents.
>
> **A1**: The scalability is a common topic for all the influence maximization papers. We would like to emphasize several points.
>
> **Theoretical assumption to reduce statistical and computational complexity**:
>
> In contrast to traditional edge-level IM papers which require maintaining all $O(N\times N)$ edges in the memory, our paper essentially assumes a low-rank structure of the transition matrices, which is able to reduce the memory consumption to $O(Nd)$, where $d$ is the feature dimension. This reduction is significant and essential for scalability.
>
> **Practically efficient implementation is possible**:
>
> For practical implementation, we improve scalability by both algorithmic modifications and hardware acceleration techniques. In our experiments, we use Monte Carlo methods, parallel computation and randomized tree search (dynamic programming) methods for faster planning, which reduces the computational complexity to $O(N)$.
>
>
> **Scale to huge graph with millions of users**:
> We thank the reviewer for the suggestion. As the first study on reinforcement learning for online adaptive IM, we have provided a theoretically-justified solution and validated it on real-world social network subgraph with ~2k nodes. This is in the same scale as previous works on online IM. For example, Vaswani et al. 2017 [1] and Wen et al. 2017 [2] evaluated their proposed algorithms on facebook subgraph with 5k nodes and 4k nodes, respectively. We agree that scaling up to huge graphs with millions of users would be an important future work for any online IM study, but this is beyond the scope of this paper.
>
> ----
>
> **Q2:** how the states are represented in the Q-learning algorithm.
>
> **A2:** In the tensor ridge regression part, the state-action pair is represented as a set of  $d$-dimensional vector $\phi_{i,k}(s,a)$. The definition of the representation is stated in Section 3. In the implementation of our algorithm, we only need to compute the representation of the current state-action pair and store $\Sigma_{t-1}$ and $B_{t-1}$ in the memory, which only requires $O(d^2)$ memory and does not scale with $N$.
>
> **Q3**: How the agent is able to efficiently explore the environment? Is it achieved by using the bonus term?
>
> **A3**: Yes, the agent efficiently explores the environment through the UCB bonus term. The definition of the bonus term is stated in Section 3. Thanks to the UCB bonus term for automatically trading-off between exploration and exploitation, we are able to prove a $\widetilde{O}(\sqrt{T})$ regret bound for our algorithm.
>
> Ps: In our experimental setting, we do not observe any hard exploration problem, and the performance of our algorithm is insensitive to the bonus term. This allows us to use approximate algorithms to calculate the bonus term, e.g., Monte-Carlo, whose computational complexity does not scale with the number of users $N$.
>
> **Q4**: What happens in the case where N is extremely large? Is it still possible to compute the covariance matrix?
>
> **A4:** With proper designs of approximation schemes, e.g. Monte-Carlo for estimating integrals & dynamic programming with pruning, the complexity of our algorithm can be reduced to O(N). This is the best computational complexity for all algorithms, since even one loop over all the nodes requires $O(N)$ complexity.
>
> We emphasize that the covariance matrix is $d \times d$, which does not scale with $N$. Therefore, computing, storing, and computing the inverse of the covariance matrix are all affordable.
>
>
> **Q5**: Matrix $A_k$ is not well defined. The sum of the elements of each row should be equal to one.
>
> **A5:** The matrix $A_k$ is well defined in Assumption 1. There are implicit requirements that the sum of the elements of each row is in $[0,1]$ to ensure every possible transition has a proper probability.
>
> **Q6**: The idea that a user has more than one chance to influence their neighbors and that of cumulative reward is not new.
>
> **A6**: We do not claim novelty over “a user has more than one chance to influence their neighbors” and “cumulative reward”. Instead, we claim novelty over the RL framework for online adaptive topic-aware influence maximization problem, together with our proposed algorithm and the regret analysis.
>
>
> [1]. Sharan Vaswani, Branislav Kveton, Zheng Wen, Mohammad Ghavamzadeh, Laks VS Lakshmanan, and Mark Schmidt. Model-independent online learning for influence maximization. In ICML, pp. 3530–3539, 2017.
>
> [2]. Zheng Wen, Branislav Kveton, Michal Valko, and Sharan Vaswani. Online influence maximization under independent cascade model with semi-bandit feedback. In NIPS, pp. 3022–3032, 2017.

---

### Official Review · Reviewer_o5dv · 2022-11-03

**Confidence:** 4
**Correctness:** 3
**Technical Novelty And Significance:** 4
**Empirical Novelty And Significance:** Not applicable
**Recommendation:** 6

**Clarity, Quality, Novelty And Reproducibility:**

This paper is generally easy to follow. It might be better to have more discussion on the regret results in Theorem 2 and 3.

**Strength And Weaknesses:**

Strength
1. This paper is the first to study the adaptive online influence maximization (OIM) problem, which is an interesting direction motivated by the recent advances in offline adaptive influence maximization.
2. The infinite-horizon discount MDP formulation of the adaptive OIM problem is reasonable.
3. The authors designed a model-based RL algorithm that only requires node-level feedback, while most previous works rely on edge-level feedback. They also proved the theoretical regret bound of the algorithm, which is the first sublinear regret bound for online adaptive influence maximization.

Weaknesses
1. The Bernoulli Independent Cascade Model relies on Assumption 3. However, IMHO, this assumption is relatively strong: it suggests that with larger $N$ and $K$, the A values should be smaller. This may not be reasonable in practice since the “influence” between two users should not be affected by the total number of users or contents.
2. In the infinite-horizon discounted MDP formulation, the action at each timestep is to activate just one user-content pair; it would be more interesting to consider the action of multiple user-content pairs per timestep.
3. I’m a bit concerned about how to obtain the user feature $x$ and content feature $\theta$ for real-world OIM problems.
4. In the problem formulation, the authors argue they focus on the asymptotic regime of large networks where $T<<N$, while in the explements, $T$ and $N$ are close (e.g., N=300, T=1000 in Fig. 1).


**Summary Of The Paper:**

This paper studied adaptive content-dependent online influence maximization where the seed nodes are sequentially activated based on real-time feedback. The authors formulated the problem as an infinite-horizon discount MDP and proposed a model-based RL algorithm that only requires node-level feedback. They proved the first sublinear regret bound for the adaptive content-dependent online influence maximization problem and validated the effectiveness of the algorithm on both synthetic and real-world data.

**Summary Of The Review:**

This paper studied a new adaptive online influence maximization problem and provided an RL algorithm with theoretical regret bound. My main concerns are Assumption 3 for the Bernoulli Independent Cascade Model and the availability of user and content features in real problems. Also, the social networks considered in the experiments are relatively small (~2000 nodes).

---

> ### Author Response · Authors · 2022-11-18
> **Thank you for your valuable review**
>
> **Q1:** The Bernoulli Independent Cascade Model relies on Assumption 3. However, IMHO, this assumption is relatively strong: it suggests that with larger $N$ and $K$, the A values should be smaller. This may not be reasonable in practice since the “influence” between two users should not be affected by the total number of users or contents.
>
> **A1:** Actually Assumption 3 reflects the fact that the meaningful discretization time step scales with the number of nodes in a network. Here is the justification for this assumption:
>
> Our Assumptions 1 and 3 actually stem from the temporal-discretization of a continuous diffusion process over a large graph, which can be formally modelled as a Hawkes process. For intuition, we can just view each user-content pair as an independent Poisson point process. In this sense, the expected waiting time for the next event (i.e., the activation/de-activation of a user) scales with $O(1/NK)$. Therefore, to get a meaningful discretization, the time step will also scales with $O(1/NK)$. This is reflected in the per time step activation matrix $A$, which linearly scales with the time step. To conclude, larger N and K results in smaller time-step, which in turn results in a small upper bound on $A$.
>
>
> **Q2**: Consider action of multiple user-content pairs per timestep.
>
> **A2**: Thank you for the suggestion! As is mentioned in **A1**, our problem formulation stems from temporal-discretization of a continuous diffusion process. Therefore, we restrict the action set to only activating one user-content pair per time step. This also simplifies the planning algorithm, as we do not need to search over an exponentially large action space. Actually, our regret bound can be extended to the case where multiple actions are allowed each timestep. For this purpose, (1) the reward $r(s,a)$ needs to be re-defined to reflect the cost of activating multiple users, and accordingly, (2) the high-probability upper bound for the reward will be $O(B / \Delta^2 \log T)$, where $B$ is the maximum number of actions allowed per timestep. Then, the final regret bound will have an additional $B$ factor.
>
> **Q3**: How to obtain the user feature and content feature for real-world OIM problems.
>
> **A3**: First, common practices to obtain user and content features in the literatures include  tensor decomposition (e.g., Vaswani et al. 2017 [1]) on the empirical transition probability tensors or apply node2vec (e.g., Wen et al. 2017 [2]) on the graph. This essentially exploits the low-rank structure of the social network. We follow the same practice using tensor decomposition to obtain user and content features on Twitter graph. Second, there are often priorly known features or some hand-designed features for users and topics. For example, one can construct demographic features for users using pre-collected information and extract semantic features for topics using language models.
>
> [1]. Sharan Vaswani, Branislav Kveton, Zheng Wen, Mohammad Ghavamzadeh, Laks VS Lakshmanan, and Mark Schmidt. Model-independent online learning for influence maximization. In ICML, pp. 3530–3539, 2017.
> [2]. Zheng Wen, Branislav Kveton, Michal Valko, and Sharan Vaswani. Online influence maximization under independent cascade model with semi-bandit feedback. In NIPS, pp. 3022–3032, 2017.
>
> **Q4:** Problem formulation considers asymptotic regime of large networks where T<<N, while in the explements, T and N are close.
>
> **A4:** Thank you for pointing out the issue! We actually wanted to say that the discretization level is smaller than $1/N$. Our theory, algorithm, and experiments *do not* rely on the magnitude between $T$ and $N$. We have removed this confusing sentence in our revised version.
>
> We hope that we have addressed your concerns on Assumption 3 and the availability of user and content features in real problems. Could you consider re-evaluating the paper and changing the score? We would like to know if you have any further questions.

---

### Decision · Program_Chairs · 2023-01-20

**Decision:**

Reject

**Justification For Why Not Higher Score:**

Simplistic model. Insufficient experimental evaluation.

**Justification For Why Not Lower Score:**

N/A

**Metareview: Summary, Strengths And Weaknesses:**

The paper studies a new adaptive online influence maximization problem and proposes a RL algorithm with sublinear regret. The reviewers all agree that the contribution is sound, however, they raise concerns regarding some of the assumptions regarding the model & the experimental evaluation (e.g., small graphs, baselines).

In my own reading of the paper, I find the model used by the authors very simplistic. As a consequence, it is unclear what would be the impact of the proposed algorithm in practice. In this context, it is rather surprising that the authors seem completely unaware of an extensive line of work in the machine learning literature (e.g., Du et al., NeurIPS 2013, JMLR 2017) which has shown that temporal point processes are strictly superior at capturing the spreading dynamics of influence and information in real networks than discrete-time models as those used by the authors.